CERN-TH-2020-132

# The Fate of Discrete 1-Form Symmetries in 6d

Fabio Apruzzi[1], Markus Dierigl[2], Ling Lin[3]

[1] *Mathematical Institute, University of Oxford,*
*Andrew-Wiles Building, Woodstock Road, Oxford, OX2 6GG, UK*
[2]*Department of Physics and Astronomy, University of Pennsylvania,*
*Philadelphia, PA 19104, USA*
[3]*CERN Theory Department, CH-1211 Geneva, Switzerland*

Recently introduced generalized global symmetries have been useful in order to understand non-perturbative aspects of quantum field theories in four and lower dimensions. In this paper we focus on 1-form symmetries of weakly coupled 6d supersymmetric gauge theories coupled to dynamical tensor multiplets. We study the consistency of global 1-form symmetries corresponding to the center of the gauge groups, or subgroups thereof, by activating their background fields, which makes the instanton density fractional. In 6d, an instanton background for a given gauge theory sources BPS strings via tadpole cancelation. The non-trivial 1-form symmetry background configurations contribute to the charge of the BPS strings. However, Dirac quantization imposes restrictions on the consistent 1-form backgrounds, since they can in general lead to and induce fractional charges, thus making (part of) the putative higher-form symmetry inconsistent. This gives explicit criteria to determine whether the discrete 1-form symmetries are realized. We implement these criteria in concrete examples originating from string compactifications. We also corroborate this by finding that a non-trivial fractional contribution is related to states which explicitly break the global 1-form symmetry appearing as massive excitations of the 6d BPS strings. For 6d theories consistently coupled to gravity, this hints at a symmetry breaking tower of states. When the fractional contributions are absent, the F-theory realization of the theories points to the gauging of the 1-form symmetry via the presence of non-trivial Mordell–Weil torsion.

# 1  Introduction

Global symmetries have always played a crucial role in the investigation of quantum field theories. In recent years the usual notion of global symmetries which act on local operators has been generalized to higher-form symmetries [1]. These generalized symmetries act on non-local operators that probe more than the local dynamics of the theory. Prominent examples with global 1-form symmetries are non-Abelian gauge theories with gauge group $G$. In the absence of matter fields that transform non-trivially under the center $Z(G)$, such theories possess a discrete global $Z(G)$ 1-form symmetry, which is generated by Gukov–Witten codmension-2 operators [2, 3], whose charged objects are one-dimensional Wilson lines. The spectrum of these extended operators is crucial in order to fully specify the theory, [4, 5]. The 1-form symmetry can be coupled to a background 2-form field, whose configurations encode possible twisted boundary conditions, i.e., non-trivial 't Hooft fluxes [6]. Furthermore, for non-Abelian gauge theories, this center 1-form symmetry can sometimes be gauged, by path-integrating over the higher-form gauge fields that couple to the charged extended objects. This affects the *global structure of the gauge group*: it results in a theory with a non-simply connected gauge group $G/Z(G)$.

Coupling the theory to background fields for various symmetries is very important and it leads to various applications. A first one is given by the computation of 't Hooft anomalies [7] for global symmetries. By coupling the theory to background fields for the global symmetries, 't Hooft anomalies are non-trivial ambiguities of the partition function under the transformation rules of these symmetries, which cannot be reabsorbed by adding local counterterms to the action. Very importantly, 't Hooft anomalies can also be mixed, meaning that the partition function is not invariant under the transformation rule of a symmetry, and that this ambiguity further depends on the background fields of another symmetry. 't Hooft anomalies can be also understood as an obstruction to gauging the symmetries involved. In the context of 0-form symmetries, anomalies have been widely utilized to study dynamics of quantum field theories (see, e.g., [8–10] for reviews).

An analogous treatment for anomalies of 1-form symmetries, in particular center symmetries of non-Abelian gauge theories, has been initiated more recently in [1, 11–15]. With the main focus on four dimensional adjoint QCD, the anomaly in question arises from the coupling of the $\theta$-angle to the instanton density $\text{Tr}(F \wedge F)$ of a $G$ gauge field configuration $F$. Since $\text{Tr}(F \wedge F)$ can develop a fractional instanton number in a non-trivial background of the $Z(G)$ 1-form symmetry, there is a mixed anomaly between the chiral symmetry acting on the adjoint fermions and the $\theta$-angle, and the $Z(G)$ 1-form symmetry. This anomaly has been very

useful for understanding the IR dynamics of SQCD theories in four dimensions [1, 12, 14–16]. Another example arises in five dimensional gauge theories. The $\mathrm{Tr}(F \wedge F)$ can couple to a $U(1)$ vector potential which can be a background or dynamical field of the theory. In both cases, if the gauge theory does not have matter that transforms non-trivially under the center, the partition function is ambiguous once the theory is coupled to a background field for the 1-form center symmetry and shifted by a large $U(1)$ (gauge) transformation of the vector potential. This leads to a trivial partition function and can be interpreted as an obstruction for activating a non-trivial background for the center 1-form symmetry in 5d; this is analyzed extensively in [17].[1]

Another application of coupling the theory to the background fields for a symmetry is to understand whether the symmetry is realized in the full theory or just approximate/emergent at certain energy scales. For instance, a symmetry can be broken by non-perturbative objects which becomes massless at some point in the moduli space of the theory. To check this it is useful to indeed look at the Dirac quantization conditions for various non-perturbative (also extended) objects of theory in the presence of non-trivial backgrounds for these symmetries. A violation of these conditions implies that the symmetry is not consistent, and therefore not a symmetry of the full theory including non-perturbative sectors.

In this work, we study discrete 1-form symmetries of six-dimensional theories with minimal, i.e., $\mathcal{N} = (1, 0)$ supersymmetry. In particular we look at 6d theories at low energies on their tensor branch (on which the scalar component of the tensor multiplets acquires a vacuum expectation value), where a weakly coupled description with an effective (pseudo[2]) Lagrangian is available. The effective theory oftentimes contains six-dimensional non-Abelian gauge sectors, where a 1-form global symmetry seems to arise. In general a 6d $\mathcal{N} = (1, 0)$ theory can be seen as an effective description of some non-trivial theory in the UV. There are several possibilities. The theory can UV-complete to a 6d superconformal field theory (SCFT) [18, 19], see [20] for a review. In this case there can only be discrete global 1-form symmetries [21], which do not have a conserved 2-form current.[3] Alternatively, the effective theory can complete to a little string theory (LST) in the UV [23–25], which allows for continuous global 1-form symmetries at low energy with current being $J_{\text{1-form}} = *_6 \mathrm{Tr}(F \wedge F)$, [21].

[1]F.A. would like to thank Pietro Benetti Genolini and Luigi Tizzano for making him aware of a similar phenomenon in 5d and for very useful discussions about their work, which have been inspirational for this paper.

[2]This is because of (anti)-self duality of the tensor multiplets, like IIB supergravity and the self-dual five-form flux $F_5$.

[3]In our paper we do not attempt to describe all 1-form symmetries of the strongly coupled SCFTs. Other non-perturbative effects can enhance or break these symmetries. On the other hand, we believe that our low energy analysis provides non-trivial information about the discussed symmetries and their fate in the UV, since as observed in [22] the global realization of symmetries can modify their tensor branch.

The third option is to couple the gauge theory to gravity, and regard it as the 6d low-energy supergravity description of a quantum theory of gravity.

In six dimensions there are 2-form tensor fields $B^i$, which can be background fields for a global $U(1)$ 1-form symmetry (which mixes non-trivially with 0-form symmetries [21]). Alternatively the $B^i$ can be dynamical if tensor multiplets are part of the low energy dynamics of the theory. The tensors couple to the instanton density of the non-Abelian gauge theory. This coupling is necessary in order to cancel reducible local gauge anomalies via (a generalization of) the Green–Schwarz–West–Sagnotti (GSWS) mechanism [26, 27]. However, it also provides a source for an ambiguity of the partition function of the effective field theory[4] involving the 1-form center symmetry.

In this paper we focus on the cases where the $B^i$ are dynamical fields, which means that their $U(1)$ 1-form symmetries are gauged. In this case, a non-trivial background flux for the gauge instanton density $\text{Tr}(F \wedge F)$ necessarily implies the addition of charged BPS strings, in order for the partition function to be non-vanishing [30], or, in other words, for tadpole cancellation. Combining this with the presence of a non-trivial $Z(G)$ background field, which can contribute fractionally to the instanton number, one encounters possible tensions with quantization of the BPS string charges. We will show in explicit examples how the 1-form symmetry background can induce fractional charge of the BPS strings present in the theory, violating therefore their Dirac quantization condition. Assuming the 6d theory to be consistent, we interpret this as an obstruction to activating the non-trivial background for the center symmetry, and consequently the absence of the discrete global 1-form symmetry. Based on this we propose a very simple criterion where the input are the data of the 6d theory and the fractional instanton densities. This in particular does not allow the gauging of global 1-form symmetry in the presence of such ambiguities, which involved the summation over non-trivial backgrounds. In principle there could be an analogous story for the dual 3-form symmetries. In this paper we choose to focus on honest $G$ gauge theories where the 1-form symmetries are meaningful. String theory constructions on the other hand should allow for both possibilities, and it should be possible to see these by computing the couplings of the 1- and 3-form symmetry background fields, [29, 31–34]. We will also attempt to give an effective field theory perspective on this.

---

[4]Many 6d theories, which come from string theory constructions, have a partition vector (rather than function) when the defect group is non-trivial, [28, 29]. In such cases one can add a free tensor with related periodicities to the theory with a partition vector and define a standard partition function. In the string theory background this can be done by specifying boundary conditions for certain fluxes. This singles out a component of the vector as the partition function [29]. In any case we expect that the ambiguity we discuss in our paper will affect the entire partition vector of 6d theory in the tensor branch (e.g., for NHCs), and that our results then apply also to these cases.

In order to corroborate that the BPS strings are a necessary ingredient that can make the discrete 1-form symmetry incompatible, we are also able to identify the states, which explicitly break the 1-form symmetry, as indeed fluctuations of the BPS strings. This explicit breaking demonstrates a posteriori why the inclusion of the non-trivial background for the 1-form $Z(G)$ symmetry is inconsistent. While these state are massive in the effective field theory, in some regime they become massless states of a strongly coupled sector, which is non-trivially charged under the weakly coupled gauge theory with center $Z(G)$. We explicitly identify the charged states by studying the elliptic genera of BPS strings of 6d theories previously analyzed in [35–39]. As expected, these states are massive in the full tensor branch, so in principle they are integrated out at low energies. Nevertheless, they impose consistency conditions concerning the coupling to the global 1-form symmetry. This can be understood as the GSWS term $B^i \wedge \mathrm{Tr}(F \wedge F)$, from which the anomaly originates, being produced in the effective action. If we assume that at some point of the tensor branch the theory has a sector consisting of non-Abelian 2-form tensor multiplet (when the scalar component has vanishing expectation value), this mechanism can be thought of as integrating out W-boson strings of a non-Abelian tensor theory. This is better defined in the circle compactification of a 6d theory to 5d. In this context the non-Abelian tensor theory reduces to a standard non-Abelian gauge theory. In the Coulomb branch of the 5d theory the W-bosons are massive, and by integrating them out certain Chern–Simons terms are produced [40, 41], which result from circle reducing the GSWS coupling in 6d. So in this way we can see that the GSWS coupling as well as the mixed anomaly, which is generated from it, are inevitably linked to the massive BPS string states.[5]

On the other hand, these states can become massless in some region of the moduli space, when the associated BPS strings are tensionless. In these regimes, the 6d models which we analyze can be still viewed as a weakly coupled gauge theory interacting with strongly coupled matter, such as the ones defined in [42]. Then the 1-form symmetry is explicitly broken by the light states coming from the tensionless strings and therefore we cannot couple the theory to its background field. Thus, the interpretation is that the violation of the Dirac quantization due to induced fractional charges on the BPS strings is a low-energy effect, which practically allows us to detect whether the theory contains symmetry breaking states becoming massless at some points in the moduli space. Using this method our findings are perfectly consistent with the results obtained via circle reduction of 6d theories to 5d KK-theories [43] and recent geometric studies of discrete M-theory fluxes related to higher-form symmetries in 5d field theories [33, 34, 44].

---

[5]These are also distinct from the hypermultiplet matter coupled to the gauge theory, which is massless everywhere on the tensor branch.

Very interestingly, for supergravity theories we find a non-trivial interplay between the obstruction to gauging the 1-form symmetries and swampland considerations [45] (see also [46, 47] for recent reviews). Namely, in a quantum theory of gravity there is strong evidence that global symmetries are absent [48–50] This is believed to hold also for discrete as well as higher-form symmetries [51–55]. Therefore, there are two possibilities for theories with non-trivial 1-form center symmetries to couple to gravity. Either, the 1-form symmetry is gauged, which means that the gauge group is $G/Z(G)$. Otherwise, the center symmetry is broken (or even absent), in which case the gauge group $G$ has a different "charge lattice" (i.e., the set of allowed matter representations) than $G/Z(G)$. By the completeness hypothesis [50], which demands that in any consistent quantum gravity theory the full charge lattice is populated, there must exist states transforming non-trivially under $Z(G)$ that explicitly break the center symmetry.[6] Note that the gauging of a higher-form symmetry induces the appearance of a dual magnetic symmetry, a magnetic 3-form symmetry in the present case in six dimensions. In a consistent theory of quantum gravity this higher-form symmetry must be broken by the presence of charged extended objects. The investigation of these states on the field theory level can be challenging and goes beyond the scope of the present paper, in string theory realizations one often can identify the magnetically charged objects with certain wrapped brane states.

We provide strong evidence that in 6d supergravity theories, a 1-form center symmetry is gauged precisely when the BPS string charges induced by the 1-form symmetry background are integer. Namely, we verify in various examples that whenever the induced charges are fractional, the BPS string carries excitations which have non-trivial center representations and thus explicitly break the 1-form symmetry. These states in turn rule out the activation of a non-trivial background for the center 1-form symmetry. Moreover, using F-theory on elliptically-fibered Calabi–Yau threefolds [57–59], we find in cases of vanishing anomaly a corresponding torsional Mordell–Weil group, which geometrically encodes a non-trivial global gauge group structure [60, 61], thus also entailing that the 1-form symmetry has been gauged.

The rest of the paper is organized as follows. In Section 2 we investigate the interplay between a shift in the dynamical tensor fields and the center 1-form symmetries in $\mathcal{N} = (1, 0)$ theories in six dimensions. A non-trivial background for the global 1-form symmetry will be excluded by charge quantization of the BPS strings, and this suggests an explicit breaking via charged states. We briefly mention the connection to five-dimensional theories derived via circle compactification, and provide the dual perspective in terms of the 3-form symmetry. In Section 3 we demonstrate the general techniques in explicit examples corresponding to supergravity models as well as SCFTs and LSTs. We comment on the different interconnections

---

[6]There can be subtle exceptions to this statement for non-Abelian discrete gauge symmetries, see [53, 56].

between the different regimes. We study the explicit breaking of the higher-form symmetries via string states in Section 4. If a non-trivial and anomaly-free global symmetry remains it can be gauged which is treated in the F-theory framework in Section 5, utilizing the structure of the Mordell–Weil torsion. We conclude and discuss some open questions in 6. Some more technical aspects can be found in the appendices.

## 2 Discrete 1-form Symmetries in 6d

In this section we first introduce some basic aspects of chiral 6d supersymmetric theories following [20]. The content of supermultiplets of 6d theories can always be expressed in terms of minimal $\mathcal{N} = (1,0)$ multiplets. For instance, $\mathcal{N} = (2,0)$ multiplets can be decomposed into $(1,0)$ components. Supersymmetry together with the representations of the little group of $SO(1,5)$, which is $Spin(4) \cong SU(2) \times SU(2)$ provide a useful organizational principle in order to list the massless supermultiplets[7]:

1. **Gravity Multiplet:** a graviton, $g_{\mu\nu}$, two gravitinos $\eta^I_\mu$ ($I = 1, 2$), and a self-dual antisymmetric tensor $\hat{B}_{\mu\nu}$, which in terms of the little group respectively are

$$(1,1) \oplus 2 \times \left(\tfrac{1}{2}, 1\right) \oplus (1,0). \tag{2.1}$$

2. **Tensor Multiplet:** a anti self-dual antisymmetric tensor $B_{\mu\nu}$, two fermions $\gamma^I$ ($I = 1, 2$), and a scalar, $\phi$. The multiplet can be written in terms of the little group as

$$(0,1) \oplus 2 \times \left(0, \tfrac{1}{2}\right) \oplus (0,0). \tag{2.2}$$

3. **Vector Multiplet:** a vector field $A_\mu$, and two fermions $\lambda^I$ ($I = 1, 2$). This multiplet can be expressed in terms of the little group as follows

$$\left(\tfrac{1}{2}, \tfrac{1}{2}\right) \oplus 2 \times \left(\tfrac{1}{2}, 0\right). \tag{2.3}$$

4. **Hypermultiplet:** two fermions $\psi^I$ ($I = 1, 2$), and four scalars, $h^\ell$

$$2 \times \left(0, \tfrac{1}{2}\right) \oplus 4 \times (0,0). \tag{2.4}$$

In this paper we will study gravitational or non-gravitational theories, in the latter the gravity multiplet is absent.

6d theories have a tensor branch moduli space when $\langle \phi^i \rangle \neq 0$. On the tensor branch an effective theory is available at low energies, and it is given in terms of the free fields listed above

---

[7]Here we present the representations of $SU(2)$ as given by the spin $s$ with $\dim(s) = 2s + 1$, i.e. $s \sim \mathbf{2s+1}$.

with certain interactions switched on. A Lagrangian for these effective theories always suffers some issue due to (anti) self-duality constraints for some tensor (similarly to the Lagrangian of IIB supergravity with self-dual $F_5$ RR flux, or chiral bosons in two-dimensional field theories). However, it is possible to write some effective Lagrangian interactions at low energies and then impose the (anti) self-duality constraints on-shell.

The general structure of a 6d theory is given by $N_T$ dynamical tensor multiplets coupled to some gauge vectors, and the formal bosonic action contains the following kinetic and interaction terms, see e.g., [62, 63],

$$S \supset 2\pi \int g_{ij} \left( -\tfrac{1}{2} d\phi^i \wedge *d\phi^j - \tfrac{1}{4} dB^i \wedge *dB^j \right) + \Omega_{ij} \left( \phi^i \wedge \tfrac{1}{4} \mathrm{Tr}(F^j \wedge *F^j) + B^i \wedge \tfrac{1}{4} \mathrm{Tr}(F^j \wedge F^j) \right) \quad (2.5)$$

where $i, j = 1, \ldots, N_T$, $F^j$ are the field strengths of the gauge vectors, and the trace is normalized such that one instanton has $\tfrac{1}{4} \int_{M_4} \mathrm{Tr}(F^2) = 1$.[8] The kinetic matrix $g_{ij}$ for the tensor sector is given by

$$g_{ij} = \phi_i \phi_j + \Omega_{ij} \,, \quad (2.6)$$

with $\phi_i = -\Omega_{ij} \phi^j$. We use conventions in which $-\Omega_{ij} \phi^i \phi^j = 2$, as in [63].

Moreover, $\Omega_{ij}$ is the Dirac pairing in the lattice of $N_T$ (anti) self-dual tensors. In 6d there are BPS strings charged with respect to $B^i$. Their tensions are $T_i \sim |\Omega_{ij} \langle \phi^j \rangle|$, and they become tensionless when $\langle \phi^j \rangle = 0$. These strings lead to massive excitations in the tensor branch, and the string charges obey the following lattice rule,

$$\langle Q^i, Q^j \rangle_{6d} = \Omega_{ij} Q^i Q^j \,, \qquad \Omega_{ij} \in \mathbb{Z} \,, \ \forall \, i, j \,. \quad (2.7)$$

The coupling between the dynamic tensors $B^i$ and $\mathrm{Tr}(F^j \wedge F^j)$,

$$\mathcal{L}_{\mathrm{GSWS}} = 2\pi \, \Omega_{ij} \left( B^i \wedge \tfrac{1}{4} \mathrm{Tr}(F^j \wedge F^j) \right), \quad (2.8)$$

plays a fundamental role in our discussion. First of all, this term is crucial in order to cancel reducible one-loop continuous gauge anomalies due to gauge transformations of the vector multiplets, that is the Green–Schwarz–West–Sagnotti mechanism [26, 27], see also [65–68].

---

[8]In order to avoid introducing extra notation we have adopted a standard convention where it looks like there is a non-trivial gauge group associated to each tensor multiplet. However, there are cases where the gauge group associated to a tensor is trivial, for these we simply have $\mathrm{Tr}(F^j \wedge F^j) = \mathrm{Tr}(F^j \wedge *F^j) = 0$. Examples of this are $(2, 0)$ tensors, which formally have $SU(1)$ gauge group associated to each tensor, or E-string theories which formally have $Sp(0)$ gauge groups for each tensor. More generally, a $G$ gauge sector in 6d is labelled by a vector $q_G$ in the tensor lattice, so that (2.8) is $2\pi \Omega_{ij} B^i q_G^j (B^i \wedge \tfrac{1}{4} \mathrm{Tr}(F^2))$ (see, e.g., [64] for a review). For tensor branch theories of SCFTs, $q_G$ typically are the basis vectors, hence the above simplified notation. However, non-trivial $q_G$ will play a role in SUGRA theories discussed in sections 3 and 5.

For instance, the one-loop anomalies can be expressed in terms of an anomaly polynomial 8-form, $I_8$, via the descent procedure. Non-reducible gauge anomalies must vanish at 1-loop, whereas reducible ones take the form

$$I_8 = \tfrac{1}{2}\Omega_{ij}\, I_4^i \wedge I_4^j\,. \tag{2.9}$$

Here,

$$I_4^j = \tfrac{1}{4}\mathrm{Tr}(F^j \wedge F^j) + (\text{global symmetry/gravity backgrounds})\,, \tag{2.10}$$

and $I_4$ receives contribution from global symmetries when we turn on backgrounds field for them. These are important in order to compute 't Hooft (mixed) anomalies. The coupling (2.8) implies that the one-loop gauge anomalies are canceled provided that the $B^i$ transform as follows under gauge transformation, $\delta$,

$$\delta B^i = -I_2, \quad \delta I_3 = dI_2, \quad I_4 = dI_3\,. \tag{2.11}$$

For example if $I_4 = \tfrac{1}{4}\mathrm{Tr}(F^2)$, $I_3 = \tfrac{1}{4}\mathrm{Tr}\big(A \wedge F - \tfrac{1}{3}A^3\big)$ and $I_2 = \tfrac{1}{4}\mathrm{Tr}(\lambda dA)$, where $\lambda$ is the gauge transformation parameter, and

$$\delta F = [F, \lambda], \qquad \delta A = d\lambda + [A, \lambda]\,. \tag{2.12}$$

The term (2.8), however, can pose a restriction to gauging 1-form symmetries that seem to be present in a low energy description of the theory.

## 2.1 Activating Backgrounds for the Discrete 1-Form Symmetries

In this section we briefly summarize properties of non-Abelian gauge theories and their generalized higher-form symmetries, by reviewing the results of [1, 14]. We will then apply these to 6d theories on their tensor branch, where we have an effective Lagrangian, (2.5), with field content given by tensor, vector and hypermultiplets.

Non-Abelian gauge theories with a simply-connected gauge group $G$ in any dimension have a discrete 1-form symmetry which corresponds to the center $Z(G)$, if matter fields which transform non-trivially under $Z(G)$ are absent [1]. In 6d the 1-form symmetry is analogous to the electric 1-form symmetry in 4d, whereas the magnetic dual is a 3-form symmetry, which again for gauge theories without matter transforming under $Z(G)$, is $Z(G)$.

The 1-form symmetry is realized by shifting the gauge potential by a flat gauge field $a$

$$A \rightarrow A + a\,, \tag{2.13}$$

where $a$ is closed with periods in $Z(G)$. In order to study ('t Hooft) anomalies of this 1-form symmetry one couples the theory to a background field, which can be formulated in terms of a

$Z(G)$-valued 2-form gauge field $C_2$. Summing over all possible background fields would result in a theory in which the center 1-form symmetry is gauged and the gauge group is modified to $G/Z(G)$, see [11]. We first consider $C_2$ a fixed non-trivial background,

$$C_2 = w_2(G/Z(G)) \ \in \ H^2(M_6, Z(G)) \,, \tag{2.14}$$

where $w_2$ is the second Stiefel–Whitney (SW) class of the quotient bundle, which encodes the obstruction to lift the $G/Z(G)$-bundle to a bundle in the simply-connected cover, i.e., a principal $G$-bundle. The instanton density,

$$I_4(G) = \tfrac{1}{4}\mathrm{Tr}(F \wedge F) \,, \tag{2.15}$$

in case of a simply-connected group is integer valued upon integration over a four-dimensional subspace in the 6d spacetime of the theory. When the bundle is twisted to the non-simply connected quotient $G/Z(G)$ due to the background $C_2$, the instanton density generically integrates to fractional values parametrized by $\alpha_G$,

$$I_4(G/Z(G)) = \alpha_G \,\mathfrak{P}(C_2) \mod \mathbb{Z} \,, \tag{2.16}$$

where $\mathfrak{P}(C_2)$ is the Pontryagin square [14,69]. If the spacetime manifold has trivial torsion the Pontryagin square can be represented by a cup product of $Z(G)$-valued 2-cocycles specified by $C_2$, $\mathfrak{P}(C_2) = C_2 \cup C_2$, [12,70].[9] For example, if $Z(G) = \mathbb{Z}_k$ with $k$ even, $\mathfrak{P}(C_2)$ is a map $H^2(M_6, \mathbb{Z}_k) \to H^4(M_6, \mathbb{Z}_{2k})$, given by

$$\mathfrak{P}: \quad C_2 \ \mapsto \ C_2' \cup C_2' \mod 2k \,, \tag{2.17}$$

where $C_2'$ is the lift of $C_2$ to an integral cocycle in $H^2(M_6, \mathbb{Z})$. For odd $k$, $\mathfrak{P}: H^2(M_6, \mathbb{Z}_k) \to H^4(M_6, \mathbb{Z}_k)$ simply coincides with the cup product. We collect the 1-form center symmetries for simply-connected groups in table 1, together with the fractional coefficients $\alpha_G$ in (2.16). Once a non-trivial background $C_2$ for the 1-form center symmetry is activated the action contains a term of the form[10]

$$S \supset 2\pi\,\Omega_{ij} \int_{M_6} B^i \wedge \alpha_G^j\, \mathfrak{P}(C_2^j) \,. \tag{2.18}$$

For $\mathcal{N} = (1,0)$ theories in six dimensions, the GSWS mechanism requires the presence of the coupling between the dynamical $B^i$ and the instanton densities of the gauge groups,

---

[9]How the Pontryagin square is defined in terms of a cup product and in which cohomology group it lives depends very much on $G$, see [11,15,70].

[10]Note that our notation is slightly abusive, since we use the wedge product and the cup product in the same expression. However, in the cases of interest the Pontryagin square also has a continuum limit as discussed e.g. in [12] and it is in this limit that we interpret the given expression, see section 2.4. If one considers a flat $[B^i] \in H^2(M_6, \mathbb{R}/\mathbb{Z})$ it can be written in terms of the cup product $[B^i] \cup \alpha_G^j\, \mathfrak{P}(C_2^j)$.

| $G$ | $Z(G)$ | $\alpha_G$ |
|:---:|:---:|:---:|
| $SU(N)$ | $\mathbb{Z}_N$ | $\frac{N-1}{2N}$ |
| $Sp(N)$ | $\mathbb{Z}_2$ | $\frac{N}{4}$ |
| $Spin(N),\ N$ odd | $\mathbb{Z}_2$ | $\frac{1}{2}$ |
| $Spin(4N+2)$ | $\mathbb{Z}_4$ | $\frac{2N+1}{8}$ |
| $Spin(4N)$ | $\mathbb{Z}_2 \times \mathbb{Z}_2$ | $\left(\frac{N}{4}, \frac{1}{2}\right)$ |
| $E_6$ | $\mathbb{Z}_3$ | $\frac{2}{3}$ |
| $E_7$ | $\mathbb{Z}_2$ | $\frac{3}{4}$ |

Table 1: Center 1-form symmetries for simply-connected gauge groups [14], and coefficients of the fractional instanton density in a non-trivial background $C_2$ [15,69]. For $Spin(4N)$ we have two coefficients $\alpha_G$ because in this case there are two contributions given by $\mathfrak{P}(C_2^{(L)} + C_2^{(R)})$ and $C_2^{(L)} \cup C_2^{(R)}$, respectively. Moreover, $F_4, G_2, E_8$ do not have center symmetries.

(2.8), in the low-energy Lagrangian. We now look at large gauge transformations of the tensor fields.[11] A consequence of this is that the GSWS coupling in the action in the presence of a fractional instanton density background will make the partition function ambiguous, and this ambiguity is not cancelled by local counterterms. This precisely looks like an anomaly. We will realize in the next subsection that BPS strings necessary for tadpole cancellation will cure the anomaly at the cost of inducing a fractional charge on them, making Dirac quantization inconsistent. The gauge transformation for $B^i$ include the following shifts,

$$B^i \to B^i + b^i \, , \tag{2.19}$$

where $b^i$ are closed 2-forms with integer periods. In this case we also assumed that the $U(1)$ 1-form gauge symmetry shifting $B^i$ and the discrete 1-form symmetry are really distinct and their transformations do not mix. The reason for this is that the $B^i$ are dynamical fields and their $U(1)$ symmetries are already gauged, whereas the discrete $Z(G)$ 1-form symmetry, when unbroken, is a global symmetry of the tensor branch theory. Note especially, that for this reason the partition function needs to be invariant under the transformations (2.19) of the dynamical tensor fields. In cases for which the $B^i$ are just background fields we cannot exclude a possible mixing a priori. We will comment on this possibility again in what follows, especially in the context of possible counterterms, but we defer a more detailed study for future work, which necessarily involves the description in terms of differential cohomology. Finally, one might wonder whether other similar anomalies are generated by the interplay between the transformation (2.11) under 0-form gauge transformations and a non-trivial background for

---

[11]From this point of view, it is convenient to view the tensor fields as non-dynamical.

the 1-form symmetry. However, these contributions will be canceled by terms generated at 1-loop by the fermionic content of the theory.

If the background $C_2^j$ is trivial the partition function is unchanged under (2.19). If, however, we activate the background fields (2.14), the gauge bundle gets twisted into a $G/Z(G)$ bundle and (2.8) generally shifts by,

$$S \to S + 2\pi\, \Omega_{ij} \int_{M_6} b^i \cup \alpha_G^j\, \mathfrak{P}(C_2^j)\,, \tag{2.20}$$

which can take fractional values when integrated[12] and lead to a phase in the partition function,

$$\mathcal{Z}[C_2^i] \to \mathcal{Z}[C_2^i] e^{2\pi i \Omega_{ij}\, \alpha_G^j}\,. \tag{2.21}$$

In particular, $\Omega_{ij}\, \alpha_G^j$ is not always an integer, signaling that the partition function might not be invariant under a (large) gauge transformation of the dynamical $B^i$ in a non-trivial background for the 1-from center symmetry. We mention in passing that there exists a 7d bulk theory, which shifts by the same anomalous phase with opposite sign on the 6d boundary. The 7d bulk theory is given by

$$S_7 = 2\pi\, \Omega_{ij} \int_{M_7} H_3^i \cup \alpha_G^j\, \mathfrak{P}(C_2^j)\,, \tag{2.22}$$

where $H_3^i \sim dB^i + \ldots$ is a 3-cochain on a 7-dimensional manifold $M_7$ which under (2.19) shifts as

$$H_3^i \to H_3^i - h^i\,, \tag{2.23}$$

$h^i$ restricts to $b^i$ on the boundary $M_6 = \partial M_7$. Moreover, when $M_7$ is closed and $H_3^i$ becomes a 3-cocycle, (2.22) is generically non-trivial, similarly to the 5d analog in [16]. This indeed points towards the presence of a mixed anomaly, meaning that there is no topological term in 6d which can completely absorb the shift.

We analyze possible local 6d counterterms which potentially could eliminate the anomalous shift, that is

$$\mathcal{A}(b_2^i, C_2^j) \equiv 2\pi\, \Omega_{ij}\, \alpha_G^j \int_{M_6} b^i \cup \mathfrak{P}(C_2^j) \tag{2.24}$$

of the partition function under the transformation (2.19) when

$$\Omega_{ij}\, \alpha_G^j \ \notin\ \mathbb{Z}\,, \tag{2.25}$$

where we do not sum over the index $j$. An obvious local counterterm is

$$\mathcal{L}_{\mathrm{CT}} = p B^i \wedge \mathfrak{P}(C_2^j)\,, \tag{2.26}$$

---

[12]In order to integrate on a closed manifold it is better to work in Euclidean signature, and therefore we would need to perform a Wick rotation. The relevant contributions will acquire the necessary factor of $i$.

which with $p = -\Omega_{ij}\alpha_G^j$ could remove the variation (2.20). However, this counterterm is not invariant under discrete 1-form symmetry transformations.

To see this, we note that a discrete 1-form gauge transformation is given by

$$C_2^j \to C_2^j + \omega_2^j\,, \tag{2.27}$$

where for gauge group $SU(N)$, $\omega_2^j = N\lambda_2^j$, and $\lambda_2^j$ is an integral 2-cochain [12]. The GSWS coupling is invariant under this transformation. For gauge group $SU(N)$ this can be seen locally by adding an extra $U(1)$ 0-form symmetry which also shift under the $\mathbb{Z}_N$ 1-form symmetry (2.27) as in [11, 12, 71], which formally extends it to a $U(N)$ gauge theory. We will briefly review this in the next section. For the other groups $G$ this can be achieved by embedding a maximal set of $SU$ subgroups into $G$, as in [15]. Under (2.27) the Pontryagin square formally shifts as

$$\mathfrak{P}(C_2^j) \to \mathfrak{P}(C_2^j) + \omega_4^j\,, \tag{2.28}$$

where $\omega_4^j$ is a non-trivial 4-form which depends on $\omega_2^j$ and $C_2^j$. For example, for $G = SU(N)$ with $N$ odd, in which case the Pontryagin square is given in terms of the cup product, we have

$$\omega_4^j = \omega_2^j \cup \omega_2^j + \omega_2^j \cup C_2^j + C_2^j \cup \omega_2^j\,, \tag{2.29}$$

where the cup product on the level of cochains is in general non-commutative.

This induces the shift $S \to S + 2\pi p \int B^i \wedge \omega_4^j$. This is better defined if we take a flat $[B^i] \in H^2(M_6, \mathbb{R}/\mathbb{Z})$ as

$$S \to S + 2\pi\Omega_{ij}\alpha_G^j \int [B^i] \cup \omega_4^j. \tag{2.30}$$

This is a shift under the symmetry transformation of the 1-form symmetry that involves the dynamical field $B^i$, which, if $\Omega_{ij}\alpha_G^j \notin \mathbb{Z}$, further does not vanish for $C_2^j = 0$. This indicates an $B^i$-operator dependent anomaly of the involved symmetries, like an ABJ-anomaly. Adding further counterterms which respect the 1-form symmetries and are of the form $\Omega_{ij}B^i \wedge P^j$, where $P^j$ are local densities invariant under the 1-form gauge transformations cannot possibly cancel the shift (2.20). Therefore, we conclude that (2.20) and (2.30) cannot simultaneously be cancelled. This might lead to the conclusion that the symmetry is violated. However, one has to further analyze whether it is possible to absorb the anomalous shift of the partition function by including physical objects in the theory. This perspective, as motivated by [30], will be studied in the section 2.2. We will find that even after the inclusion of these effects the center symmetry is broken whenever the ambiguity of the effective field theory is non-trivial.

To summarize, we have found for $\Omega_{ij}\alpha_G^j \notin \mathbb{Z}$ an ambiguity of the partition function under large gauge transformations of $B^i$ in the presence of a background field $C_2^j$ for the discrete

$Z(G^j)$ 1-form symmetry. This provides a practical way to check in which cases the center 1-form symmetries in gauge theories without matter charged under $Z(G)$ are realized in the tensor branch of the theory. Further, it is possible to consider subgroups $Z \subset \prod_j Z(G_j)$ such that the corresponding anomaly coefficient (2.25) is integer valued, implying an anomaly-free combination of the individual discrete 1-form symmetries. We will see examples of this in section 3.

Finally, it is possible to add counterterms of the type $C_2 \cup C_2 \cup C_2$ and $C_2 \cup u_4$, where $u_4$ is the gauge field for the dual 3-form (magnetic) symmetry. While these might modify the 't Hooft anomalies and the mixing of the global 1- and 3-form symmetries, they cannot possibly reabsorb the shift induced by the large gauge transformations $B \to B + b$, since we assumed that the $U(1)$ gauge symmetry acting on $B^i$ and the center symmetries of the gauge theories do not mix. Nevertheless, the inclusion of these additional topological terms might allow for interesting additional structure in the theories and affect various 't Hooft anomalies. In fact the term $C_2 \cup u_4$ is relevant in order to understand the interplay with the dual 3-form symmetry. We will partially analyze these effects in subsection 2.4 and appendix A. For more detailed analysis of these counterterms and in general of the 't Hooft anomalies we hope to come back in future work.

## 2.2 BPS Strings, Induced Charges and Dirac Quantization

We now analyze the possible consequences of the couplings (2.8) and (2.18), which generates the shifts (2.20) or (2.30), and possibly lead to ambiguities of the partition function. It might be tempting to directly claim that in the presence of this anomaly the 1-form symmetry is broken and not gaugable. However, we need to first take into account that any topologically non-trivial configuration for the instanton density would lead to a vanishing of the partition function. In fact, even before the activation of non-trivial $C_2^j$ the partition function seems to vanish for any topologically non-trivial configuration of the instanton density $I_4^j$.[13] This is due to the fact that the path integral contains an integration over flat fields $B^i$, for which

$$S_{\text{GSWS}} = 2\pi \Omega_{ij} \int_{M_6} [B^i] \cup [N_{A^j}]_{\mathbb{Z}}, \tag{2.31}$$

integrates to zero. Here, $[N_{A^j}]_{\mathbb{Z}} = [\frac{1}{4}\text{Tr}(F^j \wedge F^j)]$ denotes the cohomology class of the instanton background. This analysis fits into the framework of [30]. Taking $[I_4^j]$ for a global background, this can possibly be interpreted as the path integral measure for the dynamical field $B^i$ to acquire a charge under the global symmetries.[14] Since $[I_4^j] \supset [N_{A^j}]_{\mathbb{Z}}$ contains, however, a non-

---

[13]F.A. thanks Kantaro Ohmori for suggesting this possibility, and pointing out the reference [30].
[14]F.A. thanks Kazuya Yonekura for sharing this interpretation.

trivial configuration for a gauge field, this would make the theory inconsistent unless the effect is canceled. The cancellation proceeds by adding operators which transform with opposite charge compared to the path integral measure. In our case this is achieved by including the 6d BPS strings coupling to $B^i$. In 6d this phenomenon coincides with tadpole cancellation, which requires the inclusion of (non-perturbative BPS strings) charged objects.

More precisely, at the level of the effective field theory one has to add the operators defined in [72]

$$W_{Q_i} = \exp\left(2\pi i \int_\Sigma Q_i B^i\right). \tag{2.32}$$

This represents the electric coupling of the dynamical 2-form field $B^i$ to a string with world-volume $\Sigma$ which carries charge $Q_i = \Omega_{ij}Q^j$ defined by the string charge lattice (2.7) of the 6d theory. Therefore, when a string is present the Bianchi identity of the corresponding tensor field is modified to

$$dH^i = I_4^i + Q^i\,\sigma_4\,, \tag{2.33}$$

with $\sigma_4$ the Poincaré dual to the string worldvolume $\Sigma$. The phase in the partition function is absorbed once we demand

$$[I_4^i] = -Q^i\,\sigma_4\,, \tag{2.34}$$

i.e., the presence of strings in a topologically non-trivial instanton background. In 6d vacua coming from F-theory on Calabi–Yau threefolds the available strings are D3-branes wrapping curves in the base of the compactification manifold. The constraint (2.34) can then indeed be understood as a tadpole cancellation of the D3-brane charge in the non-trivial background. We ask now what happens when fractional values of $I_4^i$ induced by the non-trivial background for $C_2^i$ are activated. These are encoded into $\alpha_G^j$ defined in the previous subsections. The consistency condition on the induced charge dictated by the lattice of BPS string with minimal charge $q^i$ unit vector is

$$\langle q^i, \alpha_G^j\rangle = Q^i\Omega_{ij}\alpha_G^j \in \mathbb{Z} \quad \Rightarrow \quad Q_i = \Omega_{ij}\alpha_G^j \in \mathbb{Z}. \tag{2.35}$$

However, if (2.25) holds, then, due to the coupling (2.18) to the $Z(G)$ 1-form symmetry background, it seems that a BPS string (with $Q^i$ a unit vector) acquired a fractional charge, $Q_i = \Omega_{ij}\alpha_G^j \notin \mathbb{Z}$. This is not acceptable from the perspective of charge quantization, which is dictated by the string charge lattice (2.7), and for which the charges $Q_i$ need to be integers unless the fermion content on the worldvolume theory is modified, see [73].[15] In our case, since

---

[15]Alternatively, a fractional charge for a string can be interpreted as a (gauge) anomaly of the worldvolume theory, which can be sometimes canceled by worldvolume fermions. This cancellation indeed happens, for example, when orientifold-plane charges are considered, see [73].

$C_2^j$ charges the same $(0,4)$ BPS strings as $[N_{A^j}]_{\mathbb{Z}}$, the fermion content is unchanged when the background $C_2^j$ is turned on and this cancellation is not possible.[16] Therefore, we conclude that a fractional $\Omega_{ij}\,\alpha_G^j$ leads to an obstruction to turning on the non-trivial background for the discrete 1-form symmetry, which consequently is not a global symmetry of the quantum theory, and in particular cannot be gauged. In other words, the 1-form symmetries realized in 6d theories need to be compatible with the charge quantization of the strings in the spectrum.

In section 4, we support this claim by checking the explicit representation with respect to the gauge algebras of the states, which come from excitations of 6d strings necessary for tadpole cancellation. The states are massive in the full tensor branch, but there will be some regimes where the gauge theory is still weakly coupled and some of the string states are massless (even though they are non-perturbative). Exactly when $\Omega_{ij}\,\alpha_G^j \notin \mathbb{Z}$, these states will transform in representations of the gauge algebra which are not compatible with the center being a symmetry of the theory. Additionally, we will see in the next sections that in 6d examples constructed from string theory, there can be subgroups or linear combinations of $Z(G^i)$ and centers of flavor symmetries which do furnish actual global center 1-form symmetry of the theory.

## 2.3 Circle Reduction Perspective

An analogous perspective is provided when the 6d theory on the tensor branch is reduced on a circle. These 5d theories are called Kaluza–Klein (KK-)theories, and they UV complete in 6d [43, 62, 74–83]. The GSWS coupling reduces to the following Chern–Simons coupling

$$\mathcal{L}_{\mathrm{CS}} = 2\pi\, \Omega_{ij}\, A^i_{U(1)_{T_{6d}}} \wedge \tfrac{1}{4}\mathrm{Tr}(F^j \wedge F^j)\,. \tag{2.36}$$

$A^i_{U(1)_{T_{6d}}}$ is the circle reduction of $B^i$, and can be seen as a dynamical gauge field which couples to the topological currents descending from the 6d gauge groups,

$$J_T \equiv *_5 \tfrac{1}{4}\mathrm{Tr}(F^j \wedge F^j)\,. \tag{2.37}$$

The tensor branch scalars $\phi^i$ become Coulomb branch scalars associated with the $A^i_{U(1)_{T_{6d}}}$. The $A^i_{U(1)_{T_{6d}}}$ can combine with the Cartan $U(1)$ symmetries of the 6d gauge theory to form a 5d gauge theory description with an enhanced gauge algebra. The 5d gauge theory description can be useful to explicitly check candidate 1-form flavor symmetries coming from 6d 1-form and 2-form symmetries [33, 34].

---

[16]This is perhaps clearer in explicit examples of BPS strings of 6d theories coming from string constructions. An example of this can be given by the E-strings, whose charges transform in an integral lattice, even when we gauge a subgroup of $H \subset E_8$. If $Z(H) \neq 0$, a background for this symmetry does not change the worldvolume theory of the E-strings and its fermion content, but it only fractionally charges them. On the other hand we know that these strings are nevertheless consistent.

The 5d perspective is indeed useful to support that the GSWS coupling comes from integrating out some massive states. It is believed that a 6d tensor multiplet defines a non-Abelian tensor when its tensor scalar $\langle\phi^i\rangle = 0$. This reduces to a non-Abelian gauge theory in 5d, which breaks into its Cartan $U(1)^i_{T_{6d}}$ when $\langle\phi^i\rangle \neq 0$. The W-bosons of this gauge theory are in general also charged under the vector fields $A^j$ of $G^j$, which can inherit the 1-form symmetries from 6d. The Chern–Simons coupling (2.36) between $U(1)^i_{T_{6d}}$ and $G^j$ then comes by integrating out these massive W-bosons [40, 41].

We discuss a 5d $SU(3)$ gauge theory example where an analogous phenomenon happens in the Coulomb branch. For instance, a similar ambiguity is present when a W-boson is integrated out the theory is Higgsed to $SU(2) \times U(1)$. This connects the anomaly with charged massive states in appendix B.

## 2.4 Stückelberg Mechanism, Local Presentation, and Discrete 3-Form Symmetries

In this section we come back to the 6d effective theory described by tensors and vector multiplets. So far we have discussed theories with gauge group $G$. However, in general string compactifications one only has access to the Lie algebra of the gauge group with no specified global structure, unless certain boundary conditions or certain couplings are fixed [29, 31–33, 33, 34]. To see how this works field theoretically let us discuss an explicit example in the continuum limit.

We start from an $SU(N)$ gauge theory in 6d, coupled to a dynamical tensor $B$, [17]

$$\mathcal{L} \supset B \wedge \tfrac{1}{4}\mathrm{Tr}(F \wedge F)\,. \tag{2.38}$$

We continue by extending the gauge theory to $U(N)$ with connection $A'$. The background fields for the 1-form symmetry $\mathbb{Z}_N$ are given by a pair $(C_2, C)$, with $C_2$ a $\mathbb{Z}_N$ 2-form background field which is the continuous version of $C_2$ in the previous section and $C$ a $U(1)$ gauge field. They satisfy the relation $NC_2 = dC$. The action of the 1-form symmetry transformation reads,

$$
\begin{aligned}
A' &\to A' + \lambda\mathbb{I}\,, \\
C &\to C + df + N\lambda\,, \\
C_2 &\to C_2 + d\lambda\,,
\end{aligned}
\tag{2.39}
$$

---

[17]Here we discuss a simple model of one single tensor coupled to a $SU(N)$ gauge theory, which is enough for the purpose of this section. As explained in the previous section, in general 6d theories have a more complicated quiver structure and the coupling between the instanton densities and the tensors are integers different from the unit. We will see in the next section of this structure could affect the coupling to non-trivial backgrounds $C_2^j$ and the presence of global 1-form symmetries.

where $\mathbb{I}$ is the $N \times N$ unit matrix, $\lambda$ is a $U(1)$ gauge field, and a $f$ is a periodic gauge parameter which is not relevant in the following discussion. We now perform the following redefinition,

$$F \to F' - C_2\, \mathbb{I}\,, \tag{2.40}$$

The coupling (2.38) gets modified as follows

$$B \wedge \tfrac{1}{4}\mathrm{Tr}(F \wedge F) \to B \wedge \tfrac{1}{4}\mathrm{Tr}(F' \wedge F') - \tfrac{1}{N}B \wedge \mathrm{Tr}(F') \wedge dC + \tfrac{1}{2N}B \wedge dC \wedge dC\,, \tag{2.41}$$

which is invariant under (2.39). In terms of Chern classes one has

$$\tfrac{1}{4}\mathrm{Tr}(F' \wedge F') = \tfrac{1}{2}c_1(F')^2 - c_2(F')\,, \tag{2.42}$$

where $c_2(F')$ integrates to integer values.

The $U(1)$ arises naturally in string theory constructions as the center-of-mass $U(1)$ of brane stacks whose world-volume theory realizes the $U(N)$ gauge theory. However, this $U(1)$ generally acquires a mass due to a Stückelberg mechanism, which in 6d can be written in terms of a coupling $u_4 \wedge \mathrm{Tr}(F')$ with a dynamical 4-form field $u_4$ [84, 85]. In this case $u_4$ will act as a Lagrange multiplier, which integrates out the $U(1)$ gauge field. Before the inclusion of the 1-form symmetry background and the transformations (2.39), $u_4$ is just the dual 3-form symmetry background field which couples to the current $J_{U(1)}^{(4)} = *_6 \mathrm{Tr}(F')$. However, once one requires invariance with respect to (2.39), the Stückelberg term gets modified,

$$\mathcal{L}_{\mathrm{St}} = u_4 \wedge \left(\mathrm{Tr}(F') - dC\right)\,. \tag{2.43}$$

By considering $u_4$ a dynamical field, and by varying the action with respect to it, we get the constraint $\mathrm{Tr}(F') = dC = NC_2$. This tells us that the 1-form fields eliminates the 0-form $U(1)$ gauge field. Moreover, substituting this into (2.41), we get

$$\mathcal{L} \supset B \wedge \tfrac{1}{4}\big(\mathrm{Tr}(F' \wedge F') - \tfrac{1}{N}B \wedge dC \wedge dC\big) = -B \wedge c_2(F') + \tfrac{N-1}{2N}B \wedge dC \wedge dC\,, \tag{2.44}$$

where $c_2(F') = -\tfrac{1}{4}(\mathrm{Tr}(F' \wedge F') - 2\mathrm{Tr}(F') \wedge \mathrm{Tr}(F'))$, and we added and subtracted the term $\tfrac{1}{2}\mathrm{Tr}(F') \wedge \mathrm{Tr}(F')$. The second term in (2.44) is the one which leads to the anomalous phase of the partition function under large gauge transformation for $B$, which we discussed in the previous section.

Another possibility that the coupling (2.43) allows is gauging the non-anomalous part of the 1-form symmetry, (i.e., $C_2$ is not a fixed background, but becomes dynamical). This demands that $u_4$ is a background field with holonomies in $\mathbb{Z}_N$, or, in other words, a background 4-form field for the 3-form symmetry $\mathbb{Z}_N$. This also affect the 0-form $U(1)$ degrees of freedom, which

are removed by making $C_2$ dynamical in connection with the gauge transformation (2.39). This is consistent with the interpretation that one now has a $G/Z(G)$ gauge theory in 6d. By adding 1-form symmetry invariant counterterms involving $B \wedge u_4$ and $B \wedge NC_2 \wedge C_2$ or others involving $(C_2, \mathrm{Tr}(F'))$, the mixed anomaly (2.24) can be translated into a mixed anomaly for the 3-form symmetry and large gauge transformation of $B$[18]. We work out a specific choice of counterterms in appendix A, which is consistent with this view. As we saw in subsection 2.1, one could also add terms which are not invariant under the 1-form symmetry shift. This might eliminate the anomaly coming from the shift of the dynamical field $B$, however, at the same time these counterterms introduce operator $(B)$ dependent ambiguities of the partition function (2.30).

From a geometric engineering perspective in string theory both $G/Z(G)$ and $G$ should be allowed as gauge theories. The way to see this would be to compute the possible couplings between the background fields at low energy from 10/11-dimensional supergravity and by expanding brane world-volume actions. These couplings should allow for the gauging of at least one of the two symmetries, or more generally of an isotropic subgroup [29, 31–34]. In the examples we discuss in this paper we made the choice of focusing on the global 1-form symmetries of theories with group $G$.

## 3 Explicit Examples

Before discussing the presence or absence of the mixed (gauge-global symmetries) anomaly (2.24) in some explicit models, which, as discussed in section 2, does not allow the coupling to a non-trivial background $C_2$, we first briefly describe the geometric constructions of 6d $\mathcal{N} = (1,0)$ theories on their tensor branch via F-theory, [18,19].

We list here some of the most important features of consistent 6d theories on the tensor branch, as given by the geometry of the base of the torus-fibered Calabi–Yau threefolds in F-theory. The base generically looks like a set of compact curves $\Sigma_i$, which intersect according to

$$\Sigma_i \cap \Sigma_j = -\Omega_{ij} \, . \tag{3.1}$$

The tensors $B^i$ originate from the type IIB 4-form RR-field reduced on the $\Sigma_i$ (more precisely, their dual harmonic 2-forms). Gauge algebras and matter arise from intersecting 7-branes wrapping $\Sigma_i$, which are geometrized by singularities of the torus fiber in F-theory. The BPS strings come from D3-branes wrapping $\Sigma_i$. In order to summarize the base geometry we use

---

[18]Note that we trade the anomalous $\mathbb{Z}_N$ part for $C_2$ with an anomaly for the $\mathbb{Z}_N$ encoded in $u_4$. The dualization, however, involves the $U(1)$ realization of the higher-form fields.

the following notation, which in the example of the tensor branch of an SCFT is

$$[\mathfrak{g}_{\text{fl}_1}] \overset{\mathfrak{g}_1}{n_1} \cdots \underset{[\mathfrak{g}_{\text{fl}_{N_i}}]}{\overset{\mathfrak{g}_i}{n_i}} \cdots \overset{\mathfrak{g}_{N_T}}{n_{N_T}} [\mathfrak{g}_{\text{fl}_{N_T}}] . \tag{3.2}$$

Here, the compact curves $\Sigma_i$ are denoted by their negative self-intersection $n_i$, and only neighboring curves mutually intersect with intersection number 1. Recall again that $\mathfrak{g}_i$ can be trivial. The fiber can be singular also over non-compact curves which corresponds to flavor symmetries of the tensor branch theory, which we denote by $[\mathfrak{g}_{\text{fl}}]$. Such flavor symmetries are absent in supergravity models, since these are realized in F-theory on compact bases which cannot have any non-compact curves.

At last, there are three types of $\mathcal{N} = (1,0)$ theories. The intersection pairing is crucial in order to understand if the theory is a supergravity in six dimensions or if it UV-completes to a little string theory or superconformal field theory:

- The pairing of 6d superconformal field theories (SCFTs) is negative definite.

- Little string theories (LSTs) have pairings with a single zero eigenvalue, and in general there can be $N_T$ tensor multiplets with negative definite paring. Therefore the signature is $(0, N_T)$.

- The pairing of 6d supergravities has signature $(1, N_T)$, i.e., one self-dual tensor with positive signature, and $N_T$ anti-self dual tensors with negative signature. Moreover, the intersection pairing has to be unimodular [72].[19]

These are the only constraints which together with continuous anomaly cancellation conditions, see [88–90], give rise to a landscape of possible bases and tensor branches.

## 3.1 Tensor Branches of 6d Superconformal Field Theories

In this section we will demonstrate how the general procedure described above works by computing the mixed anomalies (2.24) involving the center 1-form symmetries for simple examples of 6d SCFTs on their tensor branch. Computing the discrete mixed anomalies for all 6d SCFTs coming from F-theory is far beyond the scope of this paper. Rather we select some very simple illustrative examples, and we defer a complete scan for future work. If a non-trivial induced charge $Q_i = \Omega_{ij} \alpha_G^j$ is encountered the 6d theory in the tensor branch cannot be coupled to the non-trivial background of the center 1-form, which points towards the presence of charged states as we will see in the next section.

---

[19]For more subtle anomaly constraints in the F-theory context see also [86, 87].

**Minimal 6d SCFTs:** These theories have a single tensor with string pairing $\Omega_{ij} = (n)$, which is coupled to a gauge group $G$ usually without matter. These so-called non-Higgsable clusters (NHCs) can be summarized as follows:

$$
\begin{array}{c||c|c|c|c|c|c|c}
\Sigma^2 = (-n) & -3 & -4 & -5 & -6 & -7 & -8 & -12 \\
\hline
\mathfrak{g} & \mathfrak{su}_3 & \mathfrak{so}_8 & \mathfrak{f}_4 & \mathfrak{e}_6 & \mathfrak{e}_7 + \frac{1}{2}\mathbf{56} & \mathfrak{e}_7 & \mathfrak{e}_8
\end{array}
\tag{3.3}
$$

The groups $F_4$ and $E_8$ do not have any center, and therefore there is no 1-form global symmetry. The case of $\mathfrak{e}_7 + \frac{1}{2}\mathbf{56}$ on a self-intersection $(-7)$ curve does not have any 1-form symmetry. In fact, it is broken by the presence of the massless half-hyper in the fundamental representation.

For the other cases, we can see that with these values of $n$ and the $\alpha_G$ in table 1, we have

$$
G \neq Spin(8) : \Omega_{ij}\, \alpha_G^j = n\, \alpha_G \ \in \ \mathbb{Z}\,,
\tag{3.4}
$$

and for the special case of $Spin(8)$, which has two independent contributions (see table 1), we have

$$
\Omega_{ij}\, \alpha_{Spin(8)}^{(1)} = 1\,, \quad \Omega_{ij}\, \alpha_{Spin(8)}^{(2)} = 2\,,
\tag{3.5}
$$

where the evenness of the second term as the coefficient of $C_2^{(L)} \cup C_2^{(R)}$ is necessary for consistency [70]. Therefore there is no mixed induced fractional charge on the BPS strings and the $Z(G)$ 1-form symmetries of these NHCs are not broken on the tensor branch.

**Multi-curve NHCs:** Beyond the NHCs with only a single tensor there are three clusters descending from several mutually intersecting compact curves. In the notation explained in (3.2) these are given by

$$
\overset{\mathfrak{g}_2\ \mathfrak{su}_2}{3\ \ 2}\,, \qquad\qquad \overset{\mathfrak{g}_2\ \mathfrak{su}_2}{3\ \ 2}\ 2\,, \qquad\qquad \overset{\mathfrak{su}_2\ \mathfrak{so}_7\ \mathfrak{su}_2}{2\ \ 3\ \ 2}\,.
\tag{3.6}
$$

In the first two cases the only possible 1-form center symmetry originates from the $\mathfrak{su}_2$ factors. However it is broken explicitly already at the massless level since one finds massless hypermultiplets in the representation $\frac{1}{2}(\mathbf{7},\mathbf{2}) \oplus \frac{1}{2}(\mathbf{1},\mathbf{2})$ of the $G_2 \times SU(2)$ gauge symmetry, which transforms non-trivially under the $\mathbb{Z}_2$ center of $SU(2)$. The third case is more interesting since all the involved simply-connected gauge groups deduced from the algebras have $\mathbb{Z}_2$ center symmetry. Labelling the tensor fields and gauge sectors from left to right and using the adjacency matrix given by

$$
\Omega = \begin{pmatrix} 2 & -1 & 0 \\ -1 & 3 & -1 \\ 0 & -1 & 2 \end{pmatrix}\,,
\tag{3.7}
$$

we obtain the contribution (2.18) to the action

$$S \supset 2\pi \int \left( B^1 \wedge \left( \tfrac{1}{2}\mathfrak{P}(C_2^1) - \tfrac{1}{2}\mathfrak{P}(C_2^2) \right) + B^3 \wedge \left( \tfrac{1}{2}\mathfrak{P}(C_2^3) - \tfrac{1}{2}\mathfrak{P}(C_2^2) \right) \right.$$
$$\left. + B^2 \wedge \left( \tfrac{3}{2}\mathfrak{P}(C_2^2) - \tfrac{1}{4}\mathfrak{P}(C_2^1) - \tfrac{1}{4}\mathfrak{P}(C_2^3) \right) \right), \tag{3.8}$$

for non-trivial backgrounds for all the $\mathbb{Z}_2$ 1-form symmetries parametrized by $\mathfrak{P}(C_2^j)$. We see that each individual factor has fractional contributions, which would render this coupling inconsistent with Dirac quantization for the induced charge on the BPS strings. Moreover, we will always present the result in terms of the full GSWS topological action, and not just the violation of condition (2.35) in terms of the induced charges. This is useful because sometimes we will see that certain combination of the center symmetry backgrounds will be consistent with induced charge quantization. In the above example, the breaking of each individual center factor is clear from the hypermultiplet sector since there are massless states in the representations $\tfrac{1}{2}(\mathbf{2}, \mathbf{8})$ and $\tfrac{1}{2}(\mathbf{8}, \mathbf{2})$, which transform transform non-trivially under the individual $\mathbb{Z}_2$ factors and break the 1-form symmetries explicitly. However, the diagonal $\mathbb{Z}_2^{(d)} \subset Z(SU(2) \times Spin(7) \times SU(2)) \cong \mathbb{Z}_2^3$ leaves the matter fields invariant. An $[SU(2) \times Spin(7) \times SU(2)]/\mathbb{Z}_2^{(d)}$ gauge bundle can be constructed by tensoring three $SU(2)/\mathbb{Z}_2$ bundles (where one embeds into $Spin(7)$), all with the same second SW class $C_2$ [15]. In terms of the individual background fields $C_2^i$, this effectively amounts to setting

$$C_2^1 = C_2^2 = C_2^3 = C_2 \,, \tag{3.9}$$

which reduces the contribution (3.8) to

$$S \supset 2\pi \int B^2 \wedge \mathfrak{P}(C_2) \,, \tag{3.10}$$

with integer coefficient. Thus, the diagonal $\mathbb{Z}_2$ 1-form symmetry is anomaly-free and can be coupled to a non-trivial background. This also allows the summation over non-trivial configurations of $C_2$, i.e. a gauging of the 1-form symmetry.

To summarize, this ambiguity (2.24) provides a complementary perspective which also confirms the geometric prediction about the 1-form symmetries in the 5d reduction of these theories [33].

**Minimal 6d Conformal Matter Theories:** Minimal conformal matter theories are engineered in F-theory by collisions of non-compact curves carrying Lie algebras $\mathfrak{g}_{\mathrm{fl}_L}$ and $\mathfrak{g}_{\mathrm{fl}_R}$, respectively. The tensor branch is generically described by

$$[\mathfrak{g}_{\mathrm{fl}_L}] \overset{\mathfrak{g}_1}{n_1} \cdots \overset{\mathfrak{g}_i}{n_i} \cdots \overset{\mathfrak{g}_{N_T}}{n_{N_T}} [\mathfrak{g}_{\mathrm{fl}_R}] \,. \tag{3.11}$$

For example let us consider $\mathfrak{g}_{\mathrm{fl}_L} = \mathfrak{g}_{\mathrm{fl}_R} = \mathfrak{e}_6$, the tensor branch is

$$[\mathfrak{e}_6] \ 1 \ \overset{\mathfrak{su}_3}{3} \ 1 \ [\mathfrak{e}_6] . \tag{3.12}$$

The action coupled to the 1-form center symmetry background of $SU(3)$, which is $\mathbb{Z}_3$, is

$$\tfrac{1}{2\pi} S \supset \int_{M_6} \left( 3 \, B^2 \cup \alpha_{SU(3)} \, \mathfrak{P}(C_2) - B^1 \cup \alpha_{SU(3)} \, \mathfrak{P}(C_2) - B^3 \cup \alpha_{SU(3)} \, \mathfrak{P}(C_2) \right), \tag{3.13}$$

where only the first term integrates to an integer, whereas the second and third are fractional and take values in $\mathbb{Z}_3$, since $\alpha_{SU(3)} = \tfrac{1}{3}$. Therefore the naive 1-form symmetry associated to the center of $SU(3)$ cannot be coupled to a non-trivial background due to the fractional string charge induced on the BPS strings associated to curves with self-intersection $(-1)$.

Again, our result confirms the geometric computation of [33, 34], and it is consistent with the 5d circle reduction perspective of the theory, which at low-energy is described by quiver gauge theories with (anti-)fundamental matter [91] transforming under some continuous flavor symmetry.

Another example is given by the collision of two $[\mathfrak{so}_{8+2n}]$ singularities with $n \geq 0$. The tensor branch of the theory is given by

$$[\mathfrak{so}_{8+2n}] \ \overset{\mathfrak{sp}_n}{1} \ [\mathfrak{so}_{8+2n}] , \tag{3.14}$$

where we define $\mathfrak{sp}_1 \sim \mathfrak{su}_2$. The hypermultiplet spectrum contains massless states transforming in the fundamental representation of $Sp(n)$ that explicitly break the 1-form center symmetry.

**Other examples:** Let us consider the following tensor branch,

$$\overset{\mathfrak{e}_6}{6} \ 1 \ \overset{\mathfrak{su}_3}{3} \tag{3.15}$$

Naively, there are two 1-form symmetries due to the center of $E_6$ and $SU(3)$, which do not have coupled massless matter. The topological term in the action in the tensor branch is

$$\begin{aligned}
\tfrac{1}{2\pi} S \supset \int_{M_6} & \left( 6 \, B^1 \cup \alpha_{E_6} \, \mathfrak{P}(C_2^1) - B^2 \cup \alpha_{E_6} \, \mathfrak{P}(C_2^1) \right. \\
& \left. + 3 \, B^3 \cup \alpha_{SU(3)} \, \mathfrak{P}(C_2^2) - B^2 \cup \alpha_{SU(3)} \, \mathfrak{P}(C_2^2) \right),
\end{aligned} \tag{3.16}$$

The dangerous terms which could lead to induced fractional charges are

$$\tfrac{1}{2\pi} S \supset \int_{M_6} \left( - \tfrac{2}{3} \, B^2 \cup \mathfrak{P}(C_2^1) - \tfrac{1}{3} \, B^2 \cup \mathfrak{P}(C_2^2) \right). \tag{3.17}$$

From this, we can see that while the individual centers cannot be coupled to non-trivial backgrounds, the diagonal combination $C_2^1 = C_2^2 = C_2$ leads to an integer induced charge and

is therefore consistent with Dirac quantization. So only the diagonal combination survives as a global 1-form $\mathbb{Z}_3$ symmetry.

We can do a similar analysis of the conformal matter from $\mathfrak{so}_{8+2n}$ singularities. When gauging an $\mathfrak{so}$ factor, we introduce an additional $\mathfrak{sp}$ flavor symmetry, and the tensor branch configuration is

$$[\mathfrak{sp}_n] \overset{\mathfrak{so}_{8+2n}}{4} \overset{\mathfrak{sp}_n}{1} [\mathfrak{so}_{8+2n}]. \tag{3.18}$$

The $\mathfrak{so}$ gauge theory now lives on a curve with self-intersection $(-4)$. As above the massless matter states contain fields in the representation $(\mathbf{8 + 2n, n})$, where $\mathbf{8 + 2n}$ is the vector representation of $\mathfrak{so}_{8+2n}$. These states break the entire $\mathbb{Z}_2$ center of $Sp(n)$, but are invariant under a $\mathbb{Z}_2$ subgroup of $Z(Spin(8 + 2n)) = \mathbb{Z}_4$ or $\mathbb{Z}_2 \times \mathbb{Z}_2$ for odd or even $n$. Up to integer contributions the relevant terms of the action in the presence of background fields for the center symmetries are

$$\begin{aligned}
\frac{1}{2\pi}S &\supset -\int_{M_6} B^{\mathfrak{so}} \cup \left(\tfrac{n}{4}\,\mathfrak{P}(C_2^{\mathfrak{sp}})\right) && (\text{even } n)\,, \\
\frac{1}{2\pi}S &\supset \int_{M_6} B^{\mathfrak{so}} \cup \left(\tfrac{n}{2}\,\mathfrak{P}(C_2^{\mathfrak{so}}) - \tfrac{n}{4}\,\mathfrak{P}(C_2^{\mathfrak{sp}})\right) && (\text{odd } n)\,,
\end{aligned} \tag{3.19}$$

for the tensor field from the curve with self-intersection $(-4)$. We see that this vanishes for $n$ a multiple of 4. However, there are also terms associated to the curve with self-intersection $(-1)$ given by

$$\begin{aligned}
\frac{1}{2\pi}S &\supset \int_{M_6} B^{\mathfrak{sp}} \cup \left(\tfrac{n}{4}\,\mathfrak{P}(C_2^{\mathfrak{sp}}) - \tfrac{n+4}{8}\,\mathfrak{P}(C_2^{\mathfrak{so},(L)} + C_2^{\mathfrak{so},(R)}) - \tfrac{1}{2}\,C_2^{\mathfrak{so},(L)} \cup C_2^{\mathfrak{so},(R)}\right) && (\text{even } n)\,, \\
\frac{1}{2\pi}S &\supset \int_{M_6} B^{\mathfrak{sp}} \cup \left(\tfrac{n}{4}\,\mathfrak{P}(C_2^{\mathfrak{sp}}) - \tfrac{n+4}{8}\,\mathfrak{P}(C_2^{\mathfrak{so}})\right) && (\text{odd } n)\,.
\end{aligned} \tag{3.20}$$

Note that this poses an obstruction for the 1-form backgrounds, even for the $\mathbb{Z}_2 \subset Z(Spin(8 + 2n))$ subgroup that leaves the vector representation invariant. Namely, a non-trivial background field $\widetilde{C}_2$ for this $\mathbb{Z}_2$ amounts to setting $C_2^{\mathfrak{so},(L)} = C_2^{\mathfrak{so},(R)} = \widetilde{C}_2$ for even $n$, and $\widetilde{C}_2 = 2C_2^{\mathfrak{so}}$ for odd $n$, which nevertheless leads to fractional charges of the BPS strings. One can also verify that no other linear combination, including the $\mathbb{Z}_2$ center of the $Sp$ gauge factor, is admissible.

If one allows the inclusion of center symmetries in the flavor sectors, then there is a way to cancel the fractional charges. Denoting by $C_2^{\mathfrak{sp}_f}$ and $C_2^{\mathfrak{so}_f}$ the center backgrounds of the $\mathfrak{sp}$

and $\mathfrak{so}$ flavor part, respectively, the relevant terms, for even $n$, in the action become

$$\int_{M_6} B^{\mathfrak{so}} \cup \left(\tfrac{n}{4}\,\mathfrak{P}(C_2^{\mathfrak{sp}}) + \tfrac{n}{4}\,\mathfrak{P}(C_2^{\mathfrak{sp}_f})\right)$$
$$+ B^{\mathfrak{sp}} \cup \left(\tfrac{n}{4}\,\mathfrak{P}(C_2^{\mathfrak{sp}}) - \tfrac{n+4}{8}\,\mathfrak{P}(C_2^{\mathfrak{so},(L)} + C_2^{\mathfrak{so},(R)}) - \tfrac{1}{2}\,C_2^{\mathfrak{so},(L)} \cup C_2^{\mathfrak{so},(R)}\right. \tag{3.21}$$
$$\left. - \tfrac{n+4}{8}\,\mathfrak{P}(C_2^{\mathfrak{so}_f,(L)} + C_2^{\mathfrak{so}_f,(R)}) - \tfrac{1}{2}\,C_2^{\mathfrak{so}_f,(L)} \cup C_2^{\mathfrak{so}_f,(R)}\right),$$

which would have no fractional charges if $C_2^{\mathfrak{sp}} = C_2^{\mathfrak{sp}_f} = C_2^{\mathfrak{so},(L)} = C_2^{\mathfrak{so}_f,(L)} \equiv \widetilde{C}_2$ and $C_2^{\mathfrak{so},(R)} = C_2^{\mathfrak{so}_f,(R)} = 0$, which corresponds to a "diagonal" $\mathbb{Z}_2$ subgroup with background $\widetilde{C}_2$ which leaves all matter hypermultiplets invariant.[20] For odd $n$, the relevant terms are

$$\int_{M_6} B^{\mathfrak{so}} \cup \left(\tfrac{n}{2}\mathfrak{P}(C_2^{\mathfrak{so}}) - \tfrac{n}{4}\,\mathfrak{P}(C_2^{\mathfrak{sp}}) - \tfrac{n}{4}\,\mathfrak{P}(C_2^{\mathfrak{sp}_f})\right)$$
$$+ B^{\mathfrak{sp}} \cup \left(\tfrac{n}{4}\,\mathfrak{P}(C_2^{\mathfrak{sp}}) - \tfrac{n+4}{8}\,\mathfrak{P}(C_2^{\mathfrak{so}}) - \tfrac{n+4}{8}\,\mathfrak{P}(C_2^{\mathfrak{so}_f})\right), \tag{3.22}$$

which allows for a diagonal $\mathbb{Z}_4$ with background fields aligned as $C_2^{\mathfrak{sp}} = C_2^{\mathfrak{sp}_f} = C_2^{\mathfrak{so}} = C_2^{\mathfrak{so}_f} \equiv \widetilde{C}_2$.[21]

## 3.2 Tensor Branches of 6d Supergravity Theories (SUGRAs)

Let us consider 6d supergravity models descending from F-theory on compact bases, which are Hirzebruch surfaces, $\mathbb{F}_n$. There are two distinct curve classes and the corresponding adjacency matrix is given by

$$\Omega_{ij} = \begin{pmatrix} n & -1 \\ -1 & 0 \end{pmatrix}, \tag{3.23}$$

where we focus on $n = 3, 4, 5, 6, 7, 8, 12$. Similarly to the non-compact NHCs the curve with negative self-intersection hosts a non-trivial gauge algebra given by (3.3), and there is no gauge algebra on the self-intersection 0 curve. Let us consider the models whose simply-connected version of the gauge groups have a non-trivial center symmetry. The coupling of the theory to a background for the 1-form symmetry, leads to

$$\tfrac{1}{2\pi}S \supset \int_{M_6} \left(n\,B^1 \cup \alpha_G\,\mathfrak{P}(C_2) - B^2 \cup \alpha_G\,\mathfrak{P}(C_2)\right). \tag{3.24}$$

The first term is integer-valued, whereas by evaluating $\alpha_G$ in table 1 for the groups of (3.3), we can see that the second term is always fractional, thus leading to a induced fractional charge for the BPS strings associated with the self-intersection 0 curve. Therefore, we have verified that the 1-form global symmetries of NHCs on single negative self-intersection curves are always

---

[20]Note that each $\mathfrak{so}$ factor in this case has a symmetry $(L) \leftrightarrow (R)$ which corresponds to exchanging the (co-)spinors. For simplicity, we have made a particular choice here.

[21]This corresponds to the subgroup generated by $(1,1,1,1) \in \mathbb{Z}_2 \times \mathbb{Z}_4 \times \mathbb{Z}_2 \times \mathbb{Z}_4 = Z(Sp_f(n) \times Spin(8+2n) \times Sp(n) \times Spin_f(8+2n))$.

broken if coupled to gravity. This is consistent with the conjecture that there are no global symmetry, including higher form symmetries, in a consistent theory of gravity [48,53,54].

More generally, we will consider, in section 5, also examples where gauge sectors are realized on a curve in the base $B$ of the F-theory geometry that is not a basis element for the tensor lattice. E.g., in the above example with $B = \mathbb{F}_n$, we can consider a gauge sector $G$ on a curve $[q_G] = q_G^1[v] + q_G^2[s]$, where $[v]$ and $[s]$ are the classes of the $(-n)$ and 0-curve, respectively. Then, the GWSW coupling is $2\pi\Omega_{ij}B^i \wedge q_G^j \frac{1}{4}\mathrm{Tr}(F \wedge F)$ with $\Omega_{ij}$ given in (3.23). The corresponding couplings, analogous to (3.16), in the presence of a background $C_2$ for the $Z(G)$ 1-form symmetry is then

$$S \supset 2\pi\,\Omega_{ij}\,q_G^j\,\alpha_G \int_{M_6} B^i \cup \mathfrak{P}(C_2)\,. \tag{3.25}$$

With the rest of the discussion going through straightforwardly, we arrive at the analogous condition, but with the vector $q_G^j$,

$$\Omega_{ij}\,q_G^j\,\alpha_G \in \mathbb{Z}\,, \tag{3.26}$$

for the absence of any induced fractional charges of the existing BPS strings resulting from (3.25).

## 3.3  Tensor Branches of 6d Little String Theories (LSTs)

For little string theories the matrix $\Omega_{ij}$ has a single zero eigenvalue. This implies that there is a linear combination of the currents

$$J^i = \tfrac{1}{4}\mathrm{Tr}(F^i \wedge F^i)\,, \tag{3.27}$$

which is not coupled to a dynamical 2-form field $B^j$. Consequently, the resulting theory contains a continuous global 1-form symmetry [21]. Let $v^i$ denote the null direction, i.e.,

$$\Omega_{ij}v^j = 0\,. \tag{3.28}$$

The current of the $U(1)$ 1-form symmetry is then given by

$$J = \sum_i v^i J^i\,. \tag{3.29}$$

In order to analyze the global group structure we need to take the anomalous transformations into consideration. This proceeds along the same lines as discussed above, which we will demonstrate in a simple example.

Consider a circle of $r$ curves of self-intersection $(-2)$, with the adjacency matrix given by

$$\Omega_{ij} = \begin{pmatrix} 2 & -1 & 0 & \dots & 0 & -1 \\ -1 & 2 & -1 & 0 & \dots & 0 \\ \vdots & & & & & \vdots \\ -1 & 0 & \dots & 0 & -1 & 2 \end{pmatrix}, \tag{3.30}$$

each of which hosts a $\mathfrak{su}_n$ gauge algebra, which in pictorial form is given by

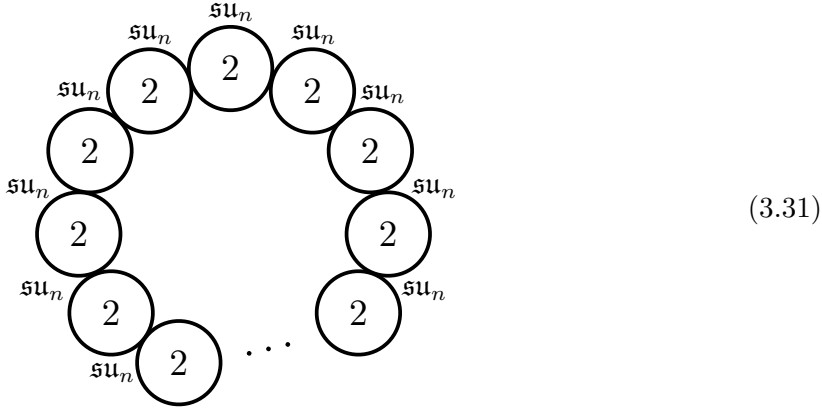

$$\tag{3.31}$$

The eigenvector with eigenvalue 0 is given by

$$v^i = \begin{pmatrix} 1 \\ \vdots \\ 1 \end{pmatrix}. \tag{3.32}$$

One finds a continuous $U(1)$ 1-form symmetry with the current

$$J = \tfrac{1}{4} \sum_i \mathrm{Tr}(F^i \wedge F^i). \tag{3.33}$$

Let us analyze what happens to the center symmetries of the $SU(n)$. The possible obstructions to switch on a non-trivial background for the center 1-form symmetries are induced by the terms

$$\begin{aligned} \tfrac{1}{2\pi} S \supset \int_{M_6} \Omega_{ij} B^i \wedge J^j &= -\int_{M_6} \left( \sum_{j=1}^r (B^{j-1} - 2B^j + B^{j+1}) \wedge J^j \right) \\ &= -\int_{M_6} \left( \sum_{i=1}^r B^i \wedge (J^{i-1} - 2J^i + J^{i+1}) \right) \end{aligned} \tag{3.34}$$

with the periodic identification $j \sim j+r$ and $i \sim i+r$. Coupling the theory to the background for the center 1-form symmetries we have

$$\tfrac{1}{2\pi} S \supset -\int_{M_6} B^i \cup (\alpha_G^{i-1} \mathfrak{P}(C_2^{i-1}) - 2\alpha_G^i \mathfrak{P}(C_2^i) + \alpha_G^{i+1} \mathfrak{P}(C_2^{i+1})). \tag{3.35}$$

We see that there is a non-trivial obstruction to activate the 1-form background in the $\mathfrak{su}_n$ factors due to fractional charges induced on the certain BPS strings. The condition to gauge part of the 1-form symmetry is that the charges induced by the 1-form symmetry background on the BPS strings are all integer. This implies that the sum of the three terms coupling to a certain tensor field $B^i$ has to have an integer quantization. In this case the bundle classes of the different gauge sectors are correlated. Note that, since these also appear in the variations of $B^{i\pm1}$ the correlation of bundle classes propagates through the full quiver. This leads to the allowed global gauge groups given by

$$G = \frac{\left(SU(n)\right)^r}{\mathbb{Z}_k} \, , \tag{3.36}$$

with $k$ a divisor of $n$. Note that this can be understood as gauging a $\mathbb{Z}_k$ subgroup of the global $U(1)$ 1-form symmetry discussed above.

In a next step we can decompactify one of the self-intersection $(-2)$ curves which leads to an A-type 6d SCFT. The corresponding adjacency matrix is obtained by deleting the $j$-th line and column. Without loss of generality we set $j = r$ and obtain

$$\Omega_{ij} = \begin{pmatrix} 2 & -1 & 0 & \ldots & 0 \\ -1 & 2 & -1 & 0 & \ldots \\ \vdots & & & & \vdots \\ 0 & \ldots & 0 & -1 & 2 \end{pmatrix} \tag{3.37}$$

leading to the quiver

$$[\mathfrak{su}_n] \overset{\mathfrak{su}_n}{2} \cdots \overset{\mathfrak{su}_n}{2} \cdots \overset{\mathfrak{su}_n}{2} [\mathfrak{su}_n] \, . \tag{3.38}$$

There is no zero eigenvalue anymore and the continuous 1-form symmetry is lost in the decompactification process. Note that now there are states transforming in the bi-fundamental representation, with one fundamental factor in the flavor symmetry. These states explicitly break the continuous 1-form symmetry. Now we can investigate the individual terms possible induced fractional charges. In the middle of the quiver the topological action coupled to the center 1-form symmetry background is given by (3.35). However, at the end of the quiver, e.g., for $i = 1$, one finds,

$$\tfrac{1}{2\pi} S \supset \int_{M_6} B^1 \cup \left( -2\alpha_G^1 \, \mathfrak{P}(C_2^1) + \alpha_G^2 \, \mathfrak{P}(C_2^2) \right) . \tag{3.39}$$

There is no combination with no fractional induced charges that only involves the gauge fields on the tensor branch of the SCFT. However, if we include a discrete background field for the center of the two $\mathfrak{su}_n$ flavor symmetries, then one can find an a combination of the center symmetries with no fractional induced charges, which essentially is the discrete remnant of the combination found in the LST example.

## 3.4  Interpolating Between Limits

The different UV embeddings of the 6d low-energy theories discussed above are of course not disconnected. In fact they often allow for continuous interpolations among them. These interpolations between theories further have nice geometric interpretations in their F-theory realizations. A variation of the scalar fields in the tensor multiplets in combination with the variation of the overall volume of the base manifold then allows to continuously connect the different regimes.

Starting with a compact base manifold we can consider the limit in which one sends the overall volume to infinity, consequently decoupling gravity, while keeping some of the curve volumes finite. If this leads to a theory with a zero eigenvalue in the intersection form of the remaining compact curves this points towards a little string sector. Since we have seen that little string theories can contain continuous higher-form symmetries this limit needs to be at infinite distance in moduli space, see, e.g., [63, 92–94], as is already guaranteed by sending the Planck mass to infinity. However, one can also take the limit in which one remains at finite base volume. This corresponds to the vanishing of the volume of a non-contractible curve configuration and is also at infinite distance in moduli space. It would be interesting to relate the 1-form symmetries in these limits to the discussion in [63, 95], where one finds an emergent dual heterotic string description.

From a general little string theory one can obtain a 6d superconformal field theory as discussed in [25], which can be understood as the further decompactification of some of the curves in the little string geometry. As discussed above the continuous 1-form symmetries have to be lost in this limit [21]. Alternatively, one can start with a supergravity theory and then contract a contractible set of curves which corresponds to the SCFT sector of the resulting theory. This point lies at finite distance in moduli space. In this description the potential discrete 1-form symmetries are either gauged or broken.

We see that the various different limits are not disconnected and their F-theory embedding allows for a fruitful geometrical interpretation. The connection to swampland criteria, especially the implications of the swampland distance conjecture, are intriguing and we wish to come back to their detailed investigation in future work. For a suggestive set of examples, given by the Hirzebruch surfaces we demonstrate the various limits and their distance in the tensor branch moduli space in appendix C.

# 4 1-Form Symmetry Breaking States

In this section we show in explicit examples that in case there is a non-trivial shift in the action under large gauge transformations of the dynamical tensor fields, there are states which explicitly break the 1-form symmetry. These states are massive in the full tensor branch and originate from the fluctuations of the BPS strings discussed in subsection 2.2. They become massless when the associated BPS strings are tensionless. There are indeed regimes where these states, even if non-perturbative, interact with the weakly coupled gauge theory with non-Abelian gauge group $G$. Moreover, when they transform non-trivially under $Z(G)$, the Wilson line operators of the non-Abelian gauge theory are screened. The BPS strings, which give rise to these modes, are indeed the ones required even at the effective field theory level for tadpole cancellation. This demonstrates from another point of view why in these cases a coupling to a non-trivial background for the 1-form symmetry is inconsistent. Therefore the perspective of this section and the one given in section 2 are fully compatible.

We will first focus on simple examples of 6d SCFTs on their tensor branch as a warm-up, and subsequently we will discuss the breaking in 6d supergravity theories. In the latter case, these states fit into a larger web of consistency conditions of quantum gravity. In particular, the absence of global symmetries means that also 1-form symmetries are either broken or gauged [48–54]. On the other hand, if a subgroup $Z$ of the center $Z(G)$ is gauged, then the gauge group is $G/Z$. The difference to $G$ manifests itself in the charge lattice, which by the completeness conjecture [50] must be fully occupied. If there is an anomaly associated to the 1-form symmetry $Z$ preventing the gauging, there must exist states in the charge lattice of $G$ which are not representations of $G/Z$. These states are not invariant under $Z$, and hence explicitly break the 1-form symmetry. We find that these states originate as excitations of dynamical strings, which also provides an interesting connection to the swampland distance conjecture [49, 63, 92–94, 96]. Here, the necessary tower of light states can sometimes be associated to the same string excitations in the effective theory.

To show this, recall that the a fractional charge induced on the BPS strings arises from coupling the theory to a non-trivial center 1-form symmetry background (2.14). In 6d theories there are indeed states which are charged under the $B^i$ as well as under the gauge potentials $A^j$. They arise from excitations of BPS strings whose tensions are $T_i \sim |\Omega_{ij}\langle\phi^j\rangle|$, and charged under the $B^i$ with charges $Q^i$ (2.7). We claim that if there exists $i, j$ such that

$$S \supset 2\pi\,\Omega_{ij}\,\alpha_G^j \int_{M_6} B^i \cup \mathfrak{P}(C_2^j) \notin \mathbb{Z}, \tag{4.1}$$

then the excitations of strings charged under $B^i$ and $G^j$ are the ones breaking the 1-form

symmetry and restrict to trivial $C_2^j$. In the F-theory context the BPS strings come from D3-branes wrapping $\Sigma_i$ in the base of the $T^2$-fibered Calabi–Yau threefold. They are electrically charged under the $B^i$ that arise from the reduction of the type IIB RR 4-form field on $\Sigma_i$. At intersections with curves $\Sigma_j$ carrying a gauge group $G^j$, there are 3-7 string states charged under $G^j$ which are precisely the states that can break the 1-form center of $G^j$.[22] Of course this is all encoded in the Dirac pairing $\Omega_{ij}$, and whether there is a non-trivial gauge group $G^j$ on $\Sigma_j$.

Let us try to understand this more concretely in some examples, starting with $(E_6, E_6)$ minimal conformal matter in the tensor branch description (3.12). In this example there is naively a $\mathbb{Z}_3$ 1-form symmetry due to the presence of the $SU(3)$ gauge theory without matter. However, the GSWS topological coupling involving the 1-form symmetry background reads

$$S \supset \int_{M_6} \left( -\tfrac{1}{3} B^1 \cup \mathfrak{P}(C_2) - \tfrac{1}{3} \cup B^3 \cup \mathfrak{P}(C_2) \right), \tag{4.2}$$

signaling that the $\mathbb{Z}_3$ backgrounds lead to an incosistency with Dirac quantization for the BPS strings of the theory. The states which break the $\mathbb{Z}_3$ are excitations of the strings charged under $B^1$ and $B^3$, both having Dirac self-pairing given by 1, and which are additionally charged under $SU(3)$. They correspond to two sets of E-strings, and they can be seen as transforming under an $E_8$ flavor symmetry, of which an $SU(3)$ subgroup has been gauged. In the F-theory setting they correspond to D3 branes wrapping $\Sigma_1$ and $\Sigma_3$, which both have self-intersection number $(-1)$. In fact, a subsector of the 2d BPS states coming from fluctuations of the string are captured by the elliptic genus of the 2d theory living on these strings [97]. In turn, the elliptic genus enters in the genus expansion of the topological strings partition function, which can be computed from the compactification geometry [35–39]. For our purposes, it is enough to analyze the genus-zero BPS states.

Before analyzing the full conformal matter example, we study the elliptic genus expansion of the minimal 6d SCFTs. These theories consist of a single curve in the base, whose self-intersection is $(-n)$, therefore they contain a single dynamical 2-form $B$ coupled to a gauge group $G$ associated to the NHC. As demonstrated in Section 3.1 there is no fractional induced charge in these cases and we expect that the center $Z(G) = \mathbb{Z}_3, \mathbb{Z}_2 \times \mathbb{Z}_2, \mathbb{Z}_3, \mathbb{Z}_2$ to be preserved when $G = SU(3), SO(8), E_6, E_7$ and $n = 3, 4, 6, 8$, respectively. Consistently, we find that the string states entering the elliptic genus all transform trivially under the center. The elliptic genus of the strings coming from D3-branes wrapping the $(-n)$ curves has been analyzed in [37, 38]. To explicitly extract the expansion it is easier to look at the limit of the elliptic

---

[22]Note while these states are massless states in the 2d world-volume description of the string on $\Sigma_i$, they cannot be in general thought of as massless particle states in the full 6d spacetime.

genus which corresponds to the Schur index of the 4d theory living on the D3 branes [98]. The first few orders of the expansion have been written in [38, Section 7 and Appendix A], and we do not repeat them here. The coefficients in this expansions correspond to representations of the Lie algebra of $G$, with respect to which the excited states transform. It can be checked that these representations are tensor products of the adjoint, which is neutral under $Z(G)$. Therefore also the components of these tensor products are neutral under $Z(G)$. For example consider $n = 3$ and $G = SU(3)$. The first representations that appear in the elliptic genus are ,

$$\mathbf{8}, \mathbf{10}, \overline{\mathbf{10}}, \mathbf{27}, \mathbf{35}, \overline{\mathbf{35}}, \mathbf{64}, \mathbf{125}, \ldots \tag{4.3}$$

These representations appear in tensor products of the adjoint representation $\mathbf{8}$ with itself:

$$\begin{aligned}
\mathbf{8}^{\otimes 4} =& 8(\mathbf{1}) \oplus 32(\mathbf{8}) \oplus 20(\mathbf{10} \oplus \overline{\mathbf{10}}) \oplus 33(\mathbf{27}) \oplus 2(\mathbf{28} \oplus \overline{\mathbf{28}}) \\
& \oplus 15(\mathbf{35} \oplus \overline{\mathbf{35}}) \oplus 12(\mathbf{64}) \oplus 3(\mathbf{81} \oplus \overline{\mathbf{81}}) \oplus \mathbf{125} \,,
\end{aligned} \tag{4.4}$$

and therefore they are all neutral under $Z(G) = \mathbb{Z}_3$. A very similar story holds for the other minimal 6d SCFTs.

Returning to the $(E_6, E_6)$ minimal conformal matter, we now find states that break the $\mathbb{Z}_3$ center symmetry of the $SU(3)$ gauge factor explicitly. This can be deduced from the fact that the E-string has an $E_8$ flavor symmetry in general. In fact, in the elliptic genus expansion [36, 39] of the E-string, the states all transform in the adjoint representation $\mathbf{248}$ of $E_8$ as well as tensor products thereof. It decomposes as

$$\mathbf{248} \to \mathbf{8} + 27 \times \mathbf{3} + 27 \times \overline{\mathbf{3}} + 78 \times \mathbf{1}. \tag{4.5}$$

with respect to an $SU(3)$ subgroup. Therefore, if we gauge such a subgroup, we see that the fundamental representation of $SU(3)$ appears which is not invariant under center transformations, and in turn breaks the $\mathbb{Z}_3$ 1-form symmetry. We find that the restriction to trivial $C_2^j$ due to the fractional charge is induced by the presence of massive states charged under the center symmetry. In this way the induced fractional charges pose a low-energy indication of the presence of charged states.

We now turn to supergravity theories, and as an illustrative example, we again discuss models engineered in F-theory on a threefold whose base is a Hirzebruch surface, $\mathbb{F}_n$, which we also analyzed in section 3. This example is very similar to the non-Higgsable cluster SCFTs, where we expect the $Z(G)$ with $G = SU(3), SO(8), E_6, E_7$ to survive due to the absence of induced fractional charges on the BPS strings. In the supergravity models, however, one has an extra dynamical 2-from tensor field in the gravity multiplet interacting with the gauge theory on the curve of negative self-intersection. On a Hirzebruch base, the additional tensor

is associated to the self-intersection $0$,[23] which leads to induced fractional charges (3.24). Because of this, the gauge group of the supergravity theory is $G$ rather than $G/Z(G)$. By the completeness hypothesis, there should hence be dynamical states charged non-trivially under $Z(G)$, which are in the charge lattice of $G$ but not that of $G/Z(G)$. Since we know for the models over $\mathbb{F}_n$ that there are no massless hypermultiplets except in the adjoint representation, these states have to originate from somewhere else. Again, these states are associated to the string from D3 branes wrapping the 0-curve. They can be thought of as critical heterotic strings at finite coupling [58, 59], whose elliptic genus can also be computed by summing two E-string elliptic genera [36]. In any case, it is clear that the visible gauge group $G = SU(3), SO(8), E_6, E_7$ is a subgroup of an $E_8$ flavor symmetry of the heterotic string. Therefore the (anti)-fundamental of $G \subset E_8$ will appear in the decomposition of $\mathbf{248}$ of the $E_8$, and thus break the $Z(G)$ 1-form symmetry explicitly.

These states appear in other contexts to guarantee the consistency of quantum gravity theories. Namely, the very same states (from D3-branes wrapping the 0-curve on Hirzebruch surfaces) have been shown [94, 95] to furnish an infinite tower of states that occupy the full charge lattice of the 0-form gauge symmetry (in the references, only $U(1)$ gauge symmetries were considered), and that these have the necessary charge-to-mass ratio to satisfy the Weak Gravity Conjecture [49]. Moreover, in accordance with the Swampland Distance Conjecture [45], these states become exponentially light as one approaches an infinite distance limit in moduli space where the 0-form symmetry becomes a global symmetry. In this limit, the notion of the 1-form symmetry becomes somewhat tenuous, as the effective description breaks down, due to the massless tower. On the other hand, the 1-form symmetry is "restored" in the limit when the 0-curve decompactifies, in which case the above string states, together with the tensor field, decouple. In this limit, also gravity decouples (see appendix C), so there is no conflict with having a global 1-form symmetry.

## 5 Gauging 1-Form Symmetries with Mordell–Weil Torsion

In the previous section, we have seen that induced fractional charges on BPS strings in the presence of a non-trivial center 1-form symmetry background is related to the existence of massive states which explicitly break the symmetry and impose the restriction $C_2^j = 0$. We have indeed seen in section 2 from the effective field theory description, that the necessary presence of BPS strings together with a topologically non-trivial fractional configuration for

---

[23]Strictly speaking, the tensor of the gravity multiplet is dual to a linear combination of the $-n$ and the 0-curve. These form a basis of tensors in the supergravity setting, so the charges associated with each must be integer.

$\mathfrak{P}(C_2)$ leads to an inconsistency with Dirac quantization. In contrast, if the induced charge are integer, the states are compatible with $C_2^j \neq 0$. Then one can contemplate the possibility of gauging the 1-form symmetry $Z$ by summing over the different non-trivial backgrounds for $C_2^j$. We approach this possibility by recalling that if a subgroup $Z$ of the full center $Z(G)$ is gauged, then the actual gauge group is $G/Z$. This perspective leads to a connection between the induced charges on the BPS strings and geometric structures in F-theory compactifications, which we will focus on in the following.

Previous works have argued that in F-theory compactifications, the global gauge group structure is encoded in the Mordell–Weil group of the elliptic fibration [60, 61]. However, strictly speaking, these arguments are only verified at the level of massless states, i.e., the massless spectrum is compatible with a non-trivial global gauge group structure $G/Z$ if the Mordell–Weil group has a torsion part $Z$.[24] The spectrum of massive states cannot be constrained a priori by the same arguments. On the other hand, as we have seen in the previous sections, a compatible massless spectrum alone is clearly not enough to guarantee a gauged center symmetry.

Moreover, this condition coming from coupling the theory to the 1-form symmetry background and the Dirac quantization of the induced charges on the BPS strings can provide a novel set of swampland constraints, if we include as a characterizing feature of an 6d supergravity theory not only the 0-form, but also the 1-form symmetries. For example, it is clear that local 0-form gauge anomalies, which only constrain massless matter, cannot detect possible obstructions from massive states to gauging a 1-form symmetry. On the other hand, a necessary condition for the 1-form symmetry $Z \subset Z(G)$ to be gauged is the absence of massless matter in non-trivial representations of $Z$, which leaves imprints on local anomaly conditions. One can in principle combine anomalies for both 0-form and 1-form gauge symmetries to constrain possible configurations of string charge lattices encoded in $\Omega_{ij}$ and non-Abelian gauge algebras to allow for a consistent 1-form center symmetry, i.e., a non-trivial global gauge group structure of a supergravity model. We hope to return to a detailed investigation of this interplay in future work.[25]

In this work, we focus on a more "streamlined" geometric criterion. Namely, we find that the Mordell–Weil group appears to automatically ensure such swampland constraints: Whenever the Mordell–Weil group of a compact F-theory model $\pi : Y \rightarrow \mathcal{B}$ has torsion

---

[24]We are not considering abelian gauge groups in this work. For these, the global structure (at the massless level) is encoded in the embedding of the free part of Mordell–Weil into the Néron–Severi group [99].

[25]In 8d supergravity models, an analogous interplay between center and continuous higher-form symmetries allows for a more direct analysis of possible global gauge groups, see [100].

$Z$,[26] there is also a consistent 1-form symmetry $Z$. This in turn means that the presence of the torsional sections should forbid not only massless hypermultiplets, but also the string states found in the previous section. This can be understood from M-/F-theory duality, which relates the 6d theory to its $S^1$-reduced 5d description in terms of M-theory on the Calabi–Yau threefold $Y$. Under this duality, the elliptic genus of 6d strings can be inferred from the topological string partition function in 5d, which in turn receives contributions from M2-branes wrapping irreducible holomorphic curves in $Y$ [103, 104]. The representation of these M2-states under the 6d gauge algebra $\mathfrak{g}$ is determined by the intersection numbers of the curves with exceptional ("Cartan") divisors $D_k$ that resolve the elliptic singularities of type $\mathfrak{g}$.

In the presence of a torsional section $\tau$, the allowed representations are restricted, due to a homology relation of the form

$$[\tau] = [\sigma_0] + \pi^{-1}(D_\mathcal{B}) + \sum_k n_k D_k \,, \tag{5.1}$$

where $D_\mathcal{B}$ is a divisor of the base determined by the intersection properties of $\tau$ and the zero section $\sigma_0$ [90, 105, 106]. Importantly, since the $n_k$ are fractional, the intersection numbers between $D_k$ and irreducible curves $C$ are restricted by the condition $[\tau] \cdot C \in \mathbb{Z}$, which is required because $[\tau]$ is an integral class [61]. By Poincaré duality, there must exist curve *classes* which fill out the full 6d charge lattice of $G/Z$ [86]. In general, these curve classes are linear combinations of curves which are not all fibral, hence they do not give massless hypermultiplets. The non-fibral irreducible curves have non-zero intersections with $\pi^{-1}(D_\mathcal{B})$, which precisely indicates that the 6d origin of their M2-brane states are the excitations of strings wrapping the $S^1$, which carry non-zero charge under the tensor dual to $D_\mathcal{B}$ [40].

For non-compact models, the situation is slightly ambiguous: Since an SCFT (and its tensor branch) is defined by local data, there can be deformations that change the global fibration without affecting the local singularity structure. These geometric deformations correspond to vacuum expectation values of operators which are irrelevant in the SCFT limit, which are known to break ordinary (0-form) global symmetries on the tensor branch. By a suitable tuning, one can make the global symmetries explicit geometrically in (nearly) all cases [107, 108]. In the context of 1-form global symmetries, we find a similar situation. Namely, whenever the SCFT does not shift under the large gauge transformations of the dynamical tensor fields in a non-trivial background for the center 1-form symmetry, we can find a complex structure deformation of the generic Weierstrass model which engineers the corresponding torsional section

---

[26] For the role of the Mordell–Weil group, and more generally the interplay between geometry and physics in F-theory, we refer to recent reviews [101, 102].

without altering the local singularity structure.

## 5.1 NHCs with Mordell–Weil Torsion

As we have seen above, non-Higgsable clusters with gauge algebra $\mathfrak{g}$ can have a consistent 1-form center $Z(G) \equiv Z$ which is broken once coupled to gravity. For these NHCs, we can always tune the corresponding elliptic fibration to have a compatible torsional Mordell–Weil group $Z$ without modifying the singularity structure on the tensor branch [22]. Globally, this tuning induces additional gauge sectors $\mathfrak{h}$, with center $Z(H) \supset Z$, on divisors that do not intersect the NHC curve(s) in the base. Their presence guarantees that the *diagonal* center $Z \subset Z(G) \times Z(H)$, represented geometrically by the Mordell–Weil torsion $Z$, is consistent with Dirac quantization of the induced charges, and thus gauged. Said differently, the geometric conditions for an elliptic threefold $Y \rightarrow \mathcal{B}$ to have Mordell–Weil torsion $Z$ automatically ensures that the corresponding supergravity theory, with a tensor spectrum specified by $\mathcal{B}$, has 0-form gauge symmetries compatible with a consistent 1-form $Z$ symmetry.

In the following, we will demonstrate this general pattern with concrete examples. For NHCs with a single tensor field we will consider the simplest "gravity completions" in terms of F-theory on Hirzebruch surfaces $\mathbb{F}_n$. We denote the homogeneous coordinates on $\mathbb{F}_n$ by $(u, v, s, t)$, with scaling relations

$$
\begin{array}{c|cccc}
 & u & v & s & t \\
\hline
\text{0-curve} & n & 0 & 1 & 1 \\
(-n)\text{-curve} & 1 & 1 & 0 & 0
\end{array}
\tag{5.2}
$$

The 0-curve has class $[s] = [t]$, and the $(-n)$-curve is $[v]$, with $[s] \cdot [v] = 1$. They form the homology basis in which the tensor pairing takes the form (3.23). The anti-canonical class is $\overline{K} = 2[v] + (n+2)[s]$. In the following, the $(+n)$-curve class $[u] = [v] + n[s]$ with $[u] \cdot [v] = 0$ will appear frequently.

**Non-Higgsable $\mathfrak{su}_3$ on $\mathbb{F}_3$**

To illustrate the observations outlined above, we focus on the NHC $\mathfrak{su}_3$ gauge algebra on a curve with $n = 3$. In this case, the generic Weierstrass model takes the form

$$
f = v^2 \tilde{f}, \quad g = v^2 \tilde{g}, \quad \Delta = v^4 (4v^2 \tilde{f}^3 + 27\tilde{g}^2).
\tag{5.3}
$$

Importantly, $[\{\tilde{g} = 0\}] = 6\overline{K} - 2[v] = 10[u]$, which means that $\{\tilde{g} = 0\}$, and hence also the residual discriminant $\{4v^2 \tilde{f}^3 + 27\tilde{g}^2 = 0\}$, do not intersect the $\mathfrak{su}_3$ divisor $\{v = 0\}$. To have a $\mathbb{Z}_3$ torsional section, $f$ and $g$ must exhibit the structure

$$
f_{\mathbb{Z}_3} = a_1 \left( \frac{a_3}{2} - \frac{a_1^3}{48} \right), \quad g_{\mathbb{Z}_3} = \frac{a_3^2}{4} - \frac{a_1^3 a_3}{24} + \frac{a_1^6}{864},
\tag{5.4}
$$

where $a_i$ are sections of the $i$-th power of the anti-canonical class $\overline{K}$ of the base [60]. On an $\mathbb{F}_3$ base, any section of these bundles has an overall factor of $v$, i.e., $a_1 = v a_1'$ and $a_3 = v a_3'$. Therefore we find

$$f_{\mathbb{Z}_3} = v^2\, a_1' \Big( \frac{a_3'}{2} - \frac{a_1'^{\,3} v^2}{48} \Big) , \quad g_{\mathbb{Z}_3} = v^2 \Big( \frac{a_3'^{\,2}}{4} - \frac{a_1'^{\,3} a_3'\, v^2}{24} + \frac{a_1'^{\,6}\, v^4}{864} \Big) ,$$
$$\Delta_{\mathbb{Z}_3} = \frac{v^4}{16} a_3'^{\,3} (27 a_3' - a_1'^{\,3}\, v^2) . \tag{5.5}$$

One can immediately verify that, because the class of $[a_3'] = 3\overline{K} - [v] = 5[u]$ has trivial intersection with $[v]$, none of the other discriminant components intersect $\{v = 0\}$. Thus, the local singularity structure over $\{v = 0\}$ remains unchanged, and still describes the non-Higgsable $\mathfrak{su}_3$, albeit with a torsion section making the $\mathbb{Z}_3$ 1-form symmetry manifest.

However, the global geometry clearly has changed drastically, most notably it now contains another $\widetilde{\mathfrak{su}}_3$ gauge algebra on $\{a_3' = 0\}$. Note that the residual discriminant intersects the $\widetilde{\mathfrak{su}}_3$ divisor at $a_3' = a_1' = 0$, however, only leading to a singularity enhancement $I_3 \to IV$ that is not accompanied by any massless hypermultiplet.

Naively, one could conclude that, since the two $\mathfrak{su}_3$ divisors do not intersect, we have two completely independent gauge factors with no massless hypermultiplets, and therefore the gauge group is $(SU(3)/\mathbb{Z}_3)^2$. However, as discussed above, a non-trivial 1-form center symmetry background of the non-Higgsable $\mathfrak{su}_3$ in the global setup induces fractional charges for the string associated to the self-intersection 0 curve $[s]$. The same reasoning applies to $\widetilde{\mathfrak{su}}_3$ on $\{a_3' = 0\}$: The class $[a_3'] = 5[u]$ enters the condition (3.26) in terms of $q_{\widetilde{\mathfrak{su}}_3}$ with $\Omega_{ij}\, q_{\widetilde{\mathfrak{su}}_3}^i\, [v]^j = -[a_3'] \cdot [v] = 0$ and $\Omega_{ij}\, q_{\widetilde{\mathfrak{su}}_3}^i\, [s]^j = -[a_3'] \cdot [s] = -5$. Therefore, the action, in the presence of a background field $C_2^{(2)}$ for the center of the $\widetilde{\mathfrak{su}}_3$, is (see (3.25))

$$\frac{1}{2\pi} S \supset - \underbrace{5\, \alpha_{\widetilde{SU}(3)}}_{\notin \mathbb{Z}} \int_{M_6} B^{[s]} \cup \mathfrak{P}(C_2^2) , \tag{5.6}$$

with $B^{[s]}$ denoting the tensor dual to the 0-curve class $[s]$. However, under the diagonal $\mathbb{Z}_3 \subset \mathbb{Z}_3 \times \mathbb{Z}_3 = Z(SU(3) \times \widetilde{SU}(3))$, i.e., after turning on a 1-form symmetry background associated to the second SW class $C_2$ of an $[SU(3) \times \widetilde{SU}(3)]/\mathbb{Z}_3$ bundle, the action reads

$$\frac{1}{2\pi} S \supset \int_{M_6} \Big( 3\, \alpha_{SU(3)}\, B^{[v]} - \big( \underbrace{\alpha_{SU(3)} + 5\alpha_{\widetilde{SU}(3)}}_{6\alpha_{SU(3)} \in \mathbb{Z}} \big)\, B^{[s]} \Big) \cup \mathfrak{P}(C_2) . \tag{5.7}$$

Hence, this diagonal $\mathbb{Z}_3$ does not induce fractional string charges.

This is in accordance with the excitations of the string from D3-branes wrapping the curve with self-intersection zero in the class $[s]$, which transform as bifundamentals under $\mathfrak{su}_3 \oplus \widetilde{\mathfrak{su}}_3$.

Since in the compact setting, the string has finite tension, these excitations are dynamic (massive) states of the theory, and break the individual $\mathbb{Z}_3$ center symmetries, but preserve the diagonal combination. In a consistent model of quantum gravity, this diagonal 1-form symmetry must therefore be gauged.

Note that this is in agreement with the geometry: there is by construction only one independent $\mathbb{Z}_3$ torsional section rather than two. More precisely, we can see that the $\mathbb{Z}_3$-section really "affects" both $\mathfrak{su}_3$ factors geometrically: The torsional section can be made explicit by the rational solutions $(x, y) = \left( \frac{(a_1' v)^2}{12}, -\frac{a_3' v}{2} \right)$ of the Weierstrass equation $y^2 = x^3 + fx + g$ with $f, g$ given as in (5.5). One can then verify that this section passes through the fiber singularities of the type IV resp. $I_3$ fiber over $v = 0$ resp. $a_3' = 0$.

To explicitly demonstrate that this leads to the typical homology relation (5.1) signaling an $[SU(3) \times \widetilde{SU}(3)]/\mathbb{Z}_3$ group structure, we have to resolve the fiber singularities over $v = 0$ and $a_3 = 0$. While we relegate the details of this resolution to appendix D, we find as a result that the homology class $[\tau]$ of the $\mathbb{Z}_3$-torsional section inside the resolved Calabi–Yau threefold satisfies

$$[\tau] = [\sigma_0] + \pi^{-1}(\overline{K}) - \tfrac{1}{3}\big(D_1^{(1)} + 2D_2^{(1)} + D_1^{(2)} + 2D_2^{(2)}\big), \tag{5.8}$$

which indeed involves the Cartan divisors $D_i^{(1/2)}$ of both $\mathfrak{su}_3$ factors.

In the limit where the volume of the curve class $[s]$ goes to infinity, also the class $[u] = [v] + [s]$ decompactifies. Therefore the string sector from D3-branes on $[s]$ as well as the $\widetilde{\mathfrak{su}}_3$ on $\{a_3' = 0\}$ become non-dynamical, leaving behind just the non-Higgsable $\mathfrak{su}_3$. One may view the 1-form center symmetry of this NHC as the remnant of the gauged $\mathbb{Z}_3$ 1-form symmetry in the compact model: Since $\widetilde{\mathfrak{su}}_3$ completely decouples, with no dynamic states charged under it, its center also decouples, so that the previously "diagonal" $\mathbb{Z}_3$ can be identified with the center of the NHC.

**Non-Higgsable $\mathfrak{so}_8$ on $\mathbb{F}_4$**

For a slightly more complicated example, we consider the $n = 4$ case with a non-Higgsable $\mathfrak{so}_8$ on $\{v = 0\}$. This gauge algebra has two independent SW classes, $C_2^{(L)}$ and $C_2^{(R)}$, associated with each factor of the $\mathbb{Z}_2^{(L)} \times \mathbb{Z}_2^{(R)}$ center. They give two contributions , $\mathfrak{P}(C_2^{(L)} + C_2^{(R)})$ and $C_2^{(L)} \cup C_2^{(R)}$, both with coefficients $\frac{1}{2}$, respectively (see table 1):

$$\begin{aligned}
\tfrac{1}{2\pi} S \supset -[v]^2 \int B^{[v]} \cup \big(\tfrac{1}{2}\mathfrak{P}(C_2^{(L)} + C_2^{(R)}) + \tfrac{1}{2}C_2^{(L)} \cup C_2^{(R)}\big) \\
= \int B^{[v]} \cup 2\big(\mathfrak{P}(C_2^{(L)} + C_2^{(R)}) + C_2^{(L)} \cup C_2^{(R)}\big).
\end{aligned} \tag{5.9}$$

Making the 1-form symmetry explicit in the elliptic fibration, we consider the generic Weierstrass model with a $\mathbb{Z}_2 \times \mathbb{Z}_2$ Mordell–Weil group [60],

$$
\begin{aligned}
f &= \tfrac{1}{3}(a_2 c_2 - a_2^2 - c_2^2) = \tfrac{1}{3}v^2(a_2' c_2' - a_2'^2 - c_2'^2)\,, \\
g &= \tfrac{1}{27}(a_2 + c_2)(2a_2 - c_2)(2c_2 - a_2) = \tfrac{1}{27}v^3(a_2' - c_2')(2a_2' - c_2')(2c_2' - a_2')\,, \\
\Delta &= -v^6 a_2'^2 c_2'^2 (a_2' - c_2')^2 \,,
\end{aligned}
\tag{5.10}
$$

where we used the fact that on the base $\mathbb{F}_4$, global sections $\omega \in \{a_2, c_2\}$ of $2\overline{K}$ factorize as $\omega = v\,\omega'$, with $[\omega'] = 3[u]$. Again, this means that the other discriminant components do not intersect $\{v = 0\}$, thus leaving the local $\mathfrak{so}_8$ NHC unchanged.

Globally, the center of $Spin(8)$ is again broken by the coupling to the tensor dual to $[s]$, which induces fractional string charges, as can be seen from

$$
\tfrac{1}{2\pi}S \supset -\underbrace{[s] \cdot [v]}_{=1} \int B^{[s]} \cup \left( \tfrac{1}{2}\mathfrak{P}(C_2^{(L)} + C_2^{(R)}) + \tfrac{1}{2}C_2^{(L)} \cup C_2^{(R)} \right).
\tag{5.11}
$$

However, in the geometry (5.10), this tensor also couples to three $\mathfrak{su}_2$ factors, which we label as follows: $\mathfrak{su}_{2,1}$ on $\{a_2' = 0\}$, $\mathfrak{su}_{2,2}$ on $\{c_2' = 0\}$, and $\mathfrak{su}_{2,3}$ on $\{a_2' - c_2' = 0\}$, each of which has curve class $3[u]$. A non-trivial 1-form center symmetry background for each of these contribute to the coupling to the tensor dual to $[s]$ as

$$
\tfrac{1}{2\pi}S \supset -3[u] \cdot [s] \int B^{[s]} \cup \tfrac{1}{4}\mathfrak{P}(C_2^{(k)}) = -\tfrac{3}{4}\int B^{[s]} \cup \mathfrak{P}(C_2^{(k)})\,, \quad k = 1, 2, 3\,.
\tag{5.12}
$$

This allows for the gauging of a "diagonal" $\mathbb{Z}_2 \times \mathbb{Z}_2$: by identifying $C_2^{(1)} = C_2^{(L)}$, $C_2^{(2)} = C_2^{(R)}$, and $C_2^{(3)} = C_2^{(L)} + C_2^{(R)}$, and using the fact that $\mathfrak{P}(C_2^{(L)} + C_2^{(R)}) = \mathfrak{P}(C_2^{(L)}) + \mathfrak{P}(C_2^{(R)}) + 2\,C_2^{(L)} \cup C_2^{(R)}$ [70], the total contribution becomes

$$
\tfrac{1}{2\pi}S \supset -\int B^{[s]} \cup \left( \left(\tfrac{1}{2} + 2\cdot\tfrac{3}{4}\right)(\mathfrak{P}(C_2^{(L)}) + \mathfrak{P}(C_2^{(R)})) + \left(1 + \tfrac{1}{2} + 2\cdot\tfrac{3}{4}\right)C_2^{(L)} \cup C_2^{(R)} \right),
\tag{5.13}
$$

which is indeed integral. This identifies $C_2^{(L)}$ and $C_2^{(R)}$ as the SW classes of a $[Spin(8) \times SU(2)_1 \times SU(2)_2 \times SU(2)_3]/[\mathbb{Z}_2^{(L)} \times \mathbb{Z}_2^{(R)}]$ bundle, where the $\mathbb{Z}_2^{(L)}$ factor is also the diagonal center of $SU(2)_1 \times SU(2)_3$, and $\mathbb{Z}_2^{(R)}$ is coupled to the diagonal center of $SU(2)_2 \times SU(2)_3$.

Note that this identification is also prescribed by the interplay between the torsional sections and the $\mathfrak{su}_2$ singularities. Namely, one can check that the Weierstrass equation $y^2 = x^3 + fx + g$, with $f, g$ given in (5.10), has three rational $\mathbb{Z}_2$-sections (all with $y = 0$), corresponding to $(1,0), (0,1), (1,1) \in \mathbb{Z}_2 \times \mathbb{Z}_2$, with $x$-coordinate

$$
x_{1,3} = \tfrac{1}{3}(2a_2' - c_2')v\,, \qquad x_{2,3} = \tfrac{1}{3}(2c_2' - a_2')v\,, \qquad x_{1,2} = -\tfrac{1}{3}(a_2' + c_2')v\,.
\tag{5.14}
$$

Clearly, they all pass through the fiber singularity at $(x, y)$ over the $\mathfrak{so}_8$ locus $\{v = 0\}$. Moreover, as indicated by the subscripts, each section $(x, y) = (x_{i,j}, 0)$ intersects the fiber singularities over two of the three $\mathfrak{su}_2$ loci with the indices $i$ and $j$.[27] This identifies the first $\mathbb{Z}_2$, generated by $(1, 0)$, as coupling the $\mathbb{Z}_2^{(L)}$ factor of $Z(Spin(8))$ with the diagonal $\mathbb{Z}_2 \subset Z(SU(2)_1 \times SU(2)_3)$, and the second $\mathbb{Z}_2$, generated by $(0, 1)$, as coupling $\mathbb{Z}_2^{(R)}$ of $Z(Spin(8))$ with the diagonal $\mathbb{Z}_2 \subset Z(SU(2)_2 \times SU(2)_3)$. Consistently, the $\mathbb{Z}_2$ generated by $(1, 1)$, which is not an independent subgroup, then couples the diagonal $\mathbb{Z}_2$ of $Z(Spin(8))$ with the diagonal of $Z(SU(2)_1 \times SU(2)_2)$.

**Multi-curve NHCs**

As discussed in section 3.1, there are three multi-curve non-Higgsable clusters. For two of them,

$$\overset{\mathfrak{g}_2}{3}\ \overset{\mathfrak{su}_2}{2} \qquad \text{and} \qquad \overset{\mathfrak{g}_2}{3}\ \overset{\mathfrak{su}_2}{2}\ 2\,, \tag{5.15}$$

the center 1-form symmetry is broken explicitly. Geometrically, this is reflected by the fact that one cannot tune a non-trivial Mordell–Weil torsion without modifying the local singularity structures [22]. In the remaining NHC,

$$\overset{\mathfrak{su}_2}{2}\ \overset{\mathfrak{so}_7}{3}\ \overset{\mathfrak{su}_2}{2}\,, \tag{5.16}$$

the diagonal $\mathbb{Z}_2 \subset Z(SU(2) \times Spin(7) \times SU(2)) \cong (\mathbb{Z}_2)^3$ is the only one that does not induce fractional string charges. Geometrically, one can indeed tune a $\mathbb{Z}_2$ torsional section in the elliptic fibration without modifying the local singularity structure [22]. Globally, we would find (on a generic base) at least another $\mathfrak{su}_2$ gauge algebra; this is necessary to cancel the fractional string charges associated to tensor from other curves that are non-compact in the local limit and hence decouple from the field theory perspective.

## 5.2 Anomaly Cancellation in Generic Torsion Models

It is amusing to consider these constraints for generic F-theory models with Mordell–Weil torsion [60]. For simplicity, we focus on models with a single $\mathbb{Z}_n$ factor. On a generic smooth base (that is, no singularity enhancement beyond the ones induced by the torsional section), these models have the following non-Abelian gauge algebras $\mathfrak{g}$ on divisor classes $D_\mathfrak{g}$, which we denote by $(\mathfrak{g}, D_\mathfrak{g})$:

$$\begin{aligned} &\mathbb{Z}_2 : \left(\mathfrak{su}_2, 4\overline{K}_\mathcal{B}\right), \quad \mathbb{Z}_3 : \left(\mathfrak{su}_3, 3\overline{K}_\mathcal{B}\right), \quad \mathbb{Z}_4 : \left(\mathfrak{su}_4, 2\overline{K}_\mathcal{B}\right), \left(\mathfrak{su}_2, \overline{K}_\mathcal{B}\right), \\ &\mathbb{Z}_5 : 2 \times \left(\mathfrak{su}_5, \overline{K}_\mathcal{B}\right), \quad \mathbb{Z}_6 : \left(\mathfrak{su}_6, \overline{K}_\mathcal{B}\right), \left(\mathfrak{su}_3, \overline{K}_\mathcal{B}\right), \left(\mathfrak{su}_2, \overline{K}_\mathcal{B}\right). \end{aligned} \tag{5.17}$$

---

[27]The $\mathfrak{su}_2$ fiber singularities are $(x, y) = \left(-\frac{c_2' v}{3}, 0\right)$ for $\mathfrak{su}_{2,1}$ over $\{a_2' = 0\}$, $(x, y) = \left(-\frac{a_2' v}{3}, 0\right)$ for $\mathfrak{su}_{2,2}$ over $\{c_2' = 0\}$, and $(x, y) = \left(\frac{c_2' v}{3}, 0\right)$ for $\mathfrak{su}_{2,3}$ over $\{a_2' = c_2'\}$.

Aside from the first two cases ($\mathbb{Z}_2$ and $\mathbb{Z}_3$), the other models all have non-minimal singularities at the intersections of the gauge divisors $D_{\mathfrak{g}}$. By blowing up these points, one finds only massless hypermultiplets in the adjoint representations of each gauge factor [109] (this holds for the first two cases without the need of blow-ups), so it appears that the full center is preserved in each case. However, only the subgroup that is isomorphic to the Mordell–Weil group $\mathbb{Z}_n$ has no obstruction to be gauged.

Namely, with a non-trivial $\mathbb{Z}_n$ 1-form symmetry background, there would be fractionally charged strings under the tensor associated to an integer divisor $D_{\mathcal{B}}$, if

$$\sum_{\mathfrak{g}} k_{\mathfrak{g}}^2 \, \alpha_G \, D_{\mathcal{B}} \cdot D_{\mathfrak{g}} \notin \mathbb{Z} \,. \tag{5.18}$$

Here, $k_{\mathfrak{g}} \in \mathbb{Z}$ denotes a possible twist of the $\mathbb{Z}_n$ embedding inside $Z(G)$, that is, the $\mathbb{Z}_n$ background field $C_2$ is $k_{\mathfrak{g}} w_2$, where $w_2$ is the second SW class of $G/Z(G)$. This immediately shows that there is no obstruction for $\mathbb{Z}_2$ and $\mathbb{Z}_3$:

$$\begin{aligned} \mathbb{Z}_2 &: k^2 \alpha_{SU(2)} D_{\mathcal{B}} \cdot D_{\mathfrak{su}_2} = \tfrac{k^2}{4} \cdot 4 D_{\mathcal{B}} \cdot \overline{K}_{\mathcal{B}} = k^2 D_{\mathcal{B}} \cdot \overline{K}_{\mathcal{B}} \in \mathbb{Z} \,, \\ \mathbb{Z}_3 &: k^2 \alpha_{SU(3)} D_{\mathcal{B}} \cdot D_{\mathfrak{su}_3} = \tfrac{k^2}{3} \cdot 3 D_{\mathcal{B}} \cdot \overline{K}_{\mathcal{B}} = k^2 D_{\mathcal{B}} \cdot \overline{K}_{\mathcal{B}} \in \mathbb{Z} \,, \end{aligned} \tag{5.19}$$

because $D_{\mathcal{B}}$ must be an integer class.

For higher $\mathfrak{su}_m$, the embedding depends on which codimension-one fiber component of the affine $\mathfrak{g}$ Dynkin diagram is intersected by the generating $\mathbb{Z}_n$ section. For $\mathbb{Z}_4$, a suitable resolution has been performed in [109], revealing that the generating $\mathbb{Z}_4$ section intersects the first non-affine $\mathfrak{su}_4$ node and the (unique) non-affine $\mathfrak{su}_2$ node. This suggests that the $\mathbb{Z}_4$ Mordell–Weil group corresponds to the "diagonal" $\mathbb{Z}_4 \subset Z(SU(4) \times SU(2)) \cong \mathbb{Z}_4 \times \mathbb{Z}_2$, generated by $(1,1)$, and therefore has $k_{\mathfrak{su}_4} = 1$. Then, there is no ambiguity given by (3.25),

$$\left( \alpha_{SU(4)} \cdot 2\overline{K}_{\mathcal{B}} + \alpha_{SU(2)} \cdot \overline{K}_{\mathcal{B}} \right) \cdot D_{\mathcal{B}} = \left( 2 \cdot \tfrac{3}{8} + \tfrac{1}{4} \right) \overline{K}_{\mathcal{B}} \cdot D_{\mathcal{B}} \in \mathbb{Z} \,, \tag{5.20}$$

allowing for a gauging of this $\mathbb{Z}_4$ 1-form symmetry, as indicated geometrically by the presence of Mordell–Weil torsion. For the $\mathbb{Z}_5$ model, the obstruction would be

$$(k_1^2 + k_2^2) \, \alpha_{SU(5)} \overline{K}_{\mathcal{B}} \cdot D_{\mathcal{B}} = \frac{2(k_1^2 + k_2^2)}{5} \overline{K}_{\mathcal{B}} \cdot D_{\mathcal{B}} \,, \tag{5.21}$$

which is trivial if $k_2 = 2k_1$. We leave a verification of this relation on threefolds for the future, but remark that, since it is a intersection of sections with codimension one fibers, the structure should be the same as for K3 surfaces which indeed satisfy a similar relationship [110]. For the $\mathbb{Z}_6$ model, there is, similar to the $\mathbb{Z}_4$ case, no ambiguity for the twist in $SU(6)$, which must be $k = 1$. Then, the obstruction is again trivial:

$$\left( \alpha_{SU(6)} + \alpha_{SU(3)} + \alpha_{SU(2)} \right) \overline{K}_{\mathcal{B}} \cdot D_{\mathcal{B}} = \left( \tfrac{5}{12} + \tfrac{1}{3} + \tfrac{1}{4} \right) \overline{K}_{\mathcal{B}} \cdot D_{\mathcal{B}} \in \mathbb{Z} \,. \tag{5.22}$$

Note that in the cases $\mathbb{Z}_4$, $\mathbb{Z}_5$, and $\mathbb{Z}_6$, the remainder of $Z(G)$, whose 1-form background field would induce fractional string charges, is explicitly broken, in accordance of our findings in Section 4. This breaking is again due to the presence of E-strings, which come from the blow-up divisors that remove the loci in $\mathcal{B}$ with non-minimal fiber singularities.

## 5.3 Global Structure of Flavor Symmetries of SCFTs

In this section, we consider the interplay of Mordell–Weil torsion and conformal matter (CM) theories, which unlike non-Higgsable clusters have 0-form flavor symmetries.

First, we revisit the example (3.15). We denote the $(-6)$-curve by $\{u = 0\}$, the $(-3)$-curve by $\{v = 0\}$, and the $(-1)$-curve by $\{e = 0\}$. Then the corresponding Weierstrass model is

$$f = \tilde{f} \, e \, u^3 v^2 \,, \quad g = \tilde{g} \, u^4 v^2 \,, \quad \Delta = u^8 v^4 (4\tilde{f}^3 e^3 u \, v^2 + 27\tilde{g}^2) \,, \tag{5.23}$$

where $\{\tilde{g}\}$ does not intersect any of the three compact curves. By setting $\tilde{f} \equiv 0$, and $\tilde{g} = a_3^2$ for suitable $a_3$, we see, first, from

$$\Delta = 27 a_3^4 u^8 v^4 \,, \tag{5.24}$$

that the local singularity structures over $\{u = 0\}$, $\{v = 0\}$ and $\{e = 0\}$ are not modified, since $\{a_3 = 0\}$ does not intersect these curves either. Second, we find in the Weierstrass equation $y^2 = x^3 + a_3^2 u^4 v^2$ two points of inflection (which are $\mathbb{Z}_3$ torsional points of opposite sign, see, e.g., [111]) $(x, y) = (0, \pm a_3 u^2 v)$, making the $\mathbb{Z}_3$ 1-form symmetry manifest.

We can now consider a decompactification of the $(-6)$- and $(-3)$-curves. In this case, the $\mathfrak{e}_6$ and $\mathfrak{su}_3$ become flavor algebras of a single E-string on $\{e = 0\}$. Since the $\mathbb{Z}_3$ 1-form symmetry was the diagonal center of these two algebras, what remains in the decompactification limit is a non-trivial global structure of the flavor symmetry, namely $[E_6 \times SU(3)]/\mathbb{Z}_3$. This is consistent with the fact that the flavor symmetry of the E-string must be a subgroup of $E_8$. Therefore, the breaking pattern of $\mathfrak{e}_8$ into maximal subalgebras come in general with non-trivial global structure, e.g., the $\mathbb{Z}_3$ quotient in case of $\mathfrak{e}_8 \to \mathfrak{e}_6 \oplus \mathfrak{su}_3$. Similarly, one also finds a compatible $\mathbb{Z}_2$-torsional section in the case of an $\mathfrak{e}_7 \oplus \mathfrak{su}_2$ collision, or a $\mathbb{Z}_5$-torsion in case of $\mathfrak{su}_5 \oplus \mathfrak{su}_5$ [22].

Another example we discussed previously was the $(E_6, E_6)$ CM, for which the anomaly is non-trivial. However, let us suppose for the moment that we gauge the two $E_6$ flavor symmetry factors. To keep the notation in (3.13), we label the tensors associated to the $\mathfrak{e}_6$ factors by $B^0$ and $B^4$. Furthermore, let us denote the self-intersection numbers of the $\mathfrak{e}_6$ divisors by $n^{(0)}$ and $n^{(4)}$, respectively. Then, we can consider the diagonal $\mathbb{Z}_3 \subset \mathbb{Z}_3^3 = Z(E_6 \times SU(3) \times E_6)$,

whose 1-form background field gives a contribution to the action as

$$\tfrac{1}{2\pi}S \supset \int_{M_6} \mathfrak{P}(C_2) \cup$$
$$\left( -n^{(0)}\alpha_{E_6}B^0 - \underbrace{(\alpha_{E_6} + \alpha_{SU(3)})}_{=1}B^1 + 3\alpha_{SU(3)}B^2 - \underbrace{(\alpha_{SU(3)} + \alpha_{E_6})}_{=1}B^3 - n^{(4)}\alpha_{E_6}B^4 \right), \qquad (5.25)$$

which is integral provided the self-intersection numbers $n^{(0)}$ and $n^{(4)}$ are multiples of three. E.g., if $n^{(0)} = n^{(4)} = -6$, in which case the two $\mathfrak{e}_6$'s are non-Higgsable, one ends up with the 6d SCFT

$$\overset{\mathfrak{e}_6}{6}\ \overset{}{1}\ \overset{\mathfrak{su}_3}{3}\ \overset{}{1}\ \overset{\mathfrak{e}_6}{6} \qquad (5.26)$$

which has a $\mathbb{Z}_3$ 1-form symmetry. Then, the conformal matter model can be thought of as the limit in which the $(-6)$-curves decompactify.

The local Weierstrass model is a transverse collision of two $\mathfrak{e}_6$ singularities over $\{u = 0\}$ and $\{v = 0\}$,

$$f = \tilde{f}\, u^3 v^3\,, \quad g = \tilde{g}\, u^4 v^4\,, \quad \Delta = u^8 v^8 (4\tilde{f}^3\, u\, v + 27\tilde{g}^2)\,, \qquad (5.27)$$

where $\tilde{g}$ does not vanish on $\{u = 0\}$ and $\{v = 0\}$. Blowing up $u = v = 0$ and any subsequent non-minimal singularities yields the above tensor branch,

$$\overset{u=0}{6}\ \overset{e_1=0}{1}\ \overset{e_2=0}{3}\ \overset{e_3=0}{1}\ \overset{v=0}{6}\,, \qquad (5.28)$$

where the upper labels denote the local coordinates. The Weierstrass model (with only minimal singularities) then is

$$f = \tilde{f}\, e_1 e_2^2 e_3\, u^3 v^3\,, \quad g = \tilde{g}\, e_2^2\, u^4 v^4\,, \quad \Delta = e_2^4 u^8 v^8 (4\tilde{f}^3 uve_1^3 e_2^2 e_3^3 + 27\tilde{g}^2)\,. \qquad (5.29)$$

Note that for consistency, prior to blowing up, $\{u = 0\}$ and $\{v = 0\}$ have self-intersection $(-4)$. Since the theory has a $\mathbb{Z}_3$ center whose background field does not induce fractional string charges, the fibration should have a local $\mathbb{Z}_3$ torsional section. To make it explicit, we can again set $\tilde{f} \equiv 0$ and $\tilde{g} = a_3^2$ for suitable $a_3$, which clearly does not change the local singularity structure, hence also not the blown-up curve configuration. In this case, the generic elliptic fiber (after base blow-up) takes the form,

$$y^2 = x^3 + a_3^2\, e_2 u^4 v^4\,, \qquad (5.30)$$

with a $\mathbb{Z}_3$-torsional section at $(x, y) = (0, a_3\, e_2 u^2 v^2)$. Note that the section passes through the fiber singularity in the $\mathfrak{e}_6$ ($u = 0$ and $v = 0$) and $\mathfrak{su}(3)$ ($e_2 = 0$) fibers, which is consistent with the fact that the unobstructed $\mathbb{Z}_3$ center is the diagonal of $Z(E_6 \times SU(3) \times E_6)$.

Decompactifying the two $\mathfrak{e}_6$ divisors, the surviving $\mathbb{Z}_3$ center is now a mix of the centers of gauge and flavor algebras. This can be interpreted as a non-trivial global structure of the flavor symmetry [15], which for the $(E_6, E_6)$ conformal matter is $[E_6 \times E_6]/\mathbb{Z}_3$. The existence of a geometric description with a $\mathbb{Z}_3$ torsional section agrees with the field theoretic computation that there are no inconsistencies for this global symmetry. Note that this statement is a priori based on a tensor branch analysis, however, we expect this global symmetry, including its non-trivial global structure, to persist at the SCFT point. It would be interesting to study this through 't Hooft anomalies with other possible higher-form symmetries of the theory.

## 6    Conclusions and Outlook

In this work we have studied discrete 1-form symmetries in 6d $\mathcal{N} = (1,0)$ theories that act as a subgroup $Z \subset Z(G)$ of the center of a non-Abelian gauge symmetry $G = \prod_j G^j$. We have focused on the interplay between this discrete higher-form center symmetry and the (gauge) $U(1)$ 1-form symmetries of (dynamical) tensor fields $B^i$ arising from the Green–Schwarz–West–Sagnotti coupling (2.8). In the presence of a background field for the center 1-form symmetry of the gauge factor $G^j$, specified by a $Z(G^j)$-valued 2-cocycle $C_2^j$, the GSWS coupling leads to a term

$$S \supset 2\pi i\, \Omega_{ij} \int_{M_6} B^i \cup \alpha_G^j\, \mathfrak{P}(C_2^j)\,. \tag{6.1}$$

If this term is fractional, it induces fractional charges on BPS strings present in the 6d theory, which are not allowed by Dirac quantization. Thus the 1-form center symmetry cannot be realized by the theory. We have also verified this a posteriori by finding charged, massive string excitations. Therefore, this provides a reliable low-energy criterion predicting when a 1-form symmetry is present or broken by non-perturbative BPS string states.

We have also studied this in explicit examples of 6d theories, varying from SCFTs on their tensor branches, to little string theories, to 6d supergravity theories. In these examples, we find a common feature: when there is an induced fractional charge due to 1-form symmetry background, there are also dynamical strings carrying (in general massive) excitations charged under the center, thus explicitly breaking it. This is reminiscent of what happens in the (partial) Coulomb branch of 5d theories, where integrating out massive W-bosons generates Chern–Simons couplings. In fact, by reducing the tensor branch theory on a circle, we find similar mixed anomalies between the center 1-form symmetries and the $U(1)$ (0-form) gauge symmetries, which originates in 5d from Chern–Simons couplings to which the GSWS coupling reduces under compactification. This agrees with recent discussion about discrete higher-form symmetries in 5d $\mathcal{N} = 1$ theories [33, 34].

Particularly interesting are theories coupled to gravity, where the above observation fits into a larger web of swampland criteria. To begin with, the absence of global symmetries in consistent quantum gravity theories [48, 51–53] implies that the center symmetries need to be either broken or gauged. On the other hand, a gauged (sub-)center $Z$ means that the gauge group is $G/Z$, which in turn has a different charge lattice than a theory with gauge group $G$. Combined with the completeness hypothesis [50], it follows that in case there is a non-trivial induced fractional charge for $Z$ obstructing its gauging, there has to be states in the charge lattice of $G$ which transform non-trivially under $Z$. In our examples, we have shown that these states are precisely the excitations of BPS strings that must exist in a consistent 6d supergravity theory. Note that the analogous states have also been shown to ensure the validity of the Weak Gravity Conjecture [94, 95] in 6d $\mathcal{N} = (1, 0)$ theories.

Furthermore, we have studied the mixed anomalies in models that arise from F-theory compactifications on elliptic Calabi–Yau threefolds with Mordell–Weil torsion $Z$. The latter is known to induce a gauge group of the form $G/Z$ [60, 61], thus imposing the gauging of a 1-form $Z$ symmetry. We have found in examples that the geometry guarantees the absence of all mixed anomalies associated with $Z$. Oftentimes, this is achieved due to non-trivial cancellations between different gauge factors of $G = \prod_j G^j$ enforced geometrically by the presence of the Mordell–Weil torsion.

Turning tables around, we can view these ambiguities as a sort of novel swampland-type constraint for theories with non-trivial gauge group structures $G/Z$, or equivalently, gauged 0- and 1-form symmetries $G$ and $Z$. Indeed, it has been previously pointed out that the geometry forbids certain combinations of $G$ and $Z$ in F-theory compactifications [109]. For example, one could have naively expected that a $Z = \mathbb{Z}_4$ center symmetry can be embedded inside an $G = SU(4)$ gauge theory. However, the generic F-theory model with $Z = \mathbb{Z}_4$ has $G = SU(4) \times SU(2)$, which, as we have shown, leads to a non-trivial cancellation for the anomalous phases associated with $\mathbb{Z}_4$. Moreover, since local (gauge) anomalies in 6d are particularly restrictive, these might conspire with the 1-form anomalies to rule out the possibility to have an $SU(4)/\mathbb{Z}_4$ consistently coupled to gravity by itself. We leave a more thorough analysis along these lines for future work.

It would also be interesting to better understand the role of Mordell–Weil torsion in local F-theory models which engineer tensor branch descriptions of SCFTs. As shown in [22], one can sometimes modify the elliptic fibration to explicitly exhibit torsion $Z$ without affecting the local singularity structure that characterizes the SCFT. This means that it is possible to freely turn on the Mordell–Weil torsion without changing the resulting SCFT, and we have shown that this is consistent with the necessary condition to have a combination consistent

with Dirac quantization of the BPS strings including the center group of flavor symmetries, which can be gauged. This is an indication that the true flavor symmetry of the SCFT is the one which is modded out by this redundancy. This geometrically allows us to predict the global structure of the flavor symmetry of the 6d SCFT.

It is known that one can use analogous geometric deformations, corresponding to vacuum expectation values of operators which are irrelevant in the UV, to make 0-form global symmetries geometrically manifest on the tensor branch of an SCFT [107, 108]. More generally, these deformations are related via dualities to non-trivial gauge backgrounds in F-theory compactifications [112–115]. Therefore, a similar interpretation for 1-form symmetries might emerge by studying the relationship to non-commuting flux backgrounds of F-theory compactifications, parallel to the discussion for 5d/4d theories from M-theory/IIB on Calabi–Yau threefolds [33, 34, 44, 116], which in turn determines the defect group structure formed by the 1-form electric and 3-form magnetic symmetries.

An interesting case to investigate would be when the there are non-dynamical tensor $B^i$ which can couple to continuous $U(1)$ 1-form 1-form symmetries. This indeed could happen for LST models as seen above, see also [21], with an interplay between this continuous $U(1)$ and the discrete 1-form symmetry coming from the center of non-Abelian 6d gauge theories (if not broken by the matter or any other state), which effectively describe the LSTs at low energies. It would be interesting to understand the obstructions to gauging these two symmetries, which in a way could be technically similar to the obstructions to activating a non-trivial background for the center 1-form symmetries encountered in this paper. For this case, it is possible that the transformations mix, leading to generalized structures for the 1-form symmetry group.

Another important aspect that we left out is the presence of $U(1)$ gauge symmetries in supergravity models, and 1-form center symmetries $Z \subset Z(G \times U(1))$ that embed non-trivially into the $U(1)$. In the absence of any non-Abelian factor $G$ and any dynamic charged states, one would expect a $U(1)$ 1-form symmetry. It would be interesting to investigate these model further, and eventually understand how they are broken or gauged.

More generally, there are also other discrete higher-form symmetries, e.g., the 2-form symmetries that form the defect group for the strings [28]. It would be interesting to study a possible gauging of these, as well as the 1-form symmetries, and understand whether they can combine in an higher group structure or not.

As we have mentioned in section 3.4 it seems that one can restore global higher-form symmetries in certain limits of the geometry. However, at these points in moduli space one also expects a tower of light states, which potentially can be related to string excitations in the effective theory [63, 95]. It is therefore of interest to study the detailed connection between

the charged string states breaking the center 1-form symmetries and the infinite distance swampland criteria. It is further plausible that the relation and mixing between different global symmetries arising at infinite distance can lead to a higher-group structure [21, 117].

## Acknowledgements

We thank Pietro Benetti Genolini, Craig Lawrie, Miguel Montero, Kantaro Ohmori, Tom Rudelius, Luigi Tizzano, Kazuya Yonekura for helpful discussions. M.D. and L.L. also thank Mirjam Cvetič and Hao Zhang for discussions and collaboration on [100]. We thank Pietro Benetti Genolini, Kantaro Ohmori, Tom Rudelius, Luigi Tizzano for reading and commenting a preliminary version of the draft. The work of F.A. is supported by the ERC Consolidator Grant number 682608 "Higgs bundles: Supersymmetric Gauge Theories and Geometry (HIGGSBNDL)". The work of M.D. is supported by the individual DFG grant DI 2527/1-1.

## A    More on Counterterms

It is possible to understand the 1-form symmetry in a more local fashion related to the continuum description of the anomalies discussed in section 2. This was done in [11, 12, 71] by embedding the $\mathfrak{su}_n$ theory into a $\mathfrak{u}_n$ theory and gauging (part of) the $U(1)$ 1-form symmetry. A similar approach can be taken for other gauge algebras by embedding the twists of the bundles into $\mathfrak{su}_2$ subalgebras as demonstrated in [15]. We have briefly summarized the continuum approach in section 2.4 applied to 6d weakly coupled theories in the case of $SU(N)$ gauge group with a $Z(SU(N)) = \mathbb{Z}_N$ 1-form center symmetry. We further elaborate on the role of counterterms.

As in section 2 we introduce a non-trivial background for the center 1-form symmetry parametrized by a 2-form $C_2$ with values in $\mathbb{Z}_N$. The continuum description introduces a continuous $U(1)$ gauge field $C$, and the relation to $C_2$ is given as[28]

$$NC_2 = dC. \tag{A.1}$$

In the following we will only work in terms of the local fields. The 1-form (background) gauge transformation acts as [11, 12]

$$C_2 \to C_2 + d\lambda, \qquad C \to C + df + N\lambda, \tag{A.2}$$

where $\lambda$ is a $U(1)$ gauge field and the scalar $f$ is related to standard gauge transformations of $C$. The $U(1)$ gauge field can also be understood as the Abelian part in an $U(N)$ bundle

---

[28]Note that we do not include the factor of $2\pi$ as in [12], since we rescale the continuum field such that, e.g., $\oint dC_1 \in \mathbb{Z}$. This also modifies some of the prefactors in the counterterms below.

parametrized by the connection $A'$, see [11, 12]. In terms of the $U(N)$ connection the relevant parts in the action, including the Stückelberg term (2.44) as discussed in section 2.4, is

$$\frac{1}{2\pi}S \supset \int_{M_6} \left( B \wedge c_2(F') - \frac{N(N-1)}{2} B \wedge C_2 \wedge C_2 + u_4 \wedge \left( \text{Tr}(F') - NC_2 \right) \right), \qquad \text{(A.3)}$$

where we expressed everything in terms of $C_2$. We can add a counterterms of the type

$$\frac{1}{2\pi}\Delta S = \int_{M_6} -pB \wedge \left( \frac{1}{N}u_4 - C_2 \wedge C_2 \right), \qquad \text{(A.4)}$$

where $p$ is an integer coefficient.[29] This term is invariant under (A.2) provided that $u_4$ shifts under the 1-form symmetry as follows,

$$u_4 \to u_4 + 2N d\lambda \wedge C_2 + N d\lambda \wedge d\lambda. \qquad \text{(A.5)}$$

Evaluating the equation of motion for $C_2$, one finds

$$u_4 = -(N-1)B \wedge C_2 + 2\frac{p}{N}B \wedge C_2 + \frac{1}{N}dC_3. \qquad \text{(A.6)}$$

By fixing the value of $p = \frac{N(N-1)}{2}$ we can get rid of the term proportional to $B \wedge C_2$. This implies that $dC_3$ is integer valued and $u_4 = \frac{1}{N}dC_3$. Plugging this back into the topological action (A.3) above, one can eliminate $C_2$ and find

$$\frac{1}{2\pi}S \supset B \wedge c_2(F') + \frac{1}{N}dC_3 \wedge \text{tr}(F') - \frac{(N-1)}{2N} dC_3 \wedge B. \qquad \text{(A.7)}$$

We have seen that by fixing the gauge invariant counterterm (A.4) and gauging the 1-form symmetry by making $C_2$ dynamical, the gauge group becomes $SU(N)/\mathbb{Z}_N$, and we have arrived at a formulation where $u_4$ is the background field for the 3-form symmetry valued in $\mathbb{Z}_N$.

Applying this to a general 6d theory in the tensor branch, we still have an anomalous phase coming from the shift of the dynamical $B$, but now mixing with the background field for the 3-form symmetry $u_4$,

$$\mathcal{A}(b_2^i, u_4^j) \equiv 2\pi \, \Omega_{ij} \, \alpha_G^j \int_{M_6} b^i \cup u_4^j. \qquad \text{(A.8)}$$

where $u_4$ can be viewed as a 4-cocycle valued in $\mathbb{Z}_N$. This implies now that it is not consistent to couple activate a non-trivial background for $u_4^j$. Accordingly, this suggests the presence of massive magnetically charges states. It is an interesting question to find their explicit realizations string theory setups.

---

[29] We can also try to add something which is not invariant under the 1-form symmetry shift. This might eliminate the anomaly coming from the shift of the dynamical field $B$, however at the same time these counterterms introduce operator $B$ dependent ambiguities of the partition function, which are both ABJ and mixed anomalies. In this work we chose counterterms such that the operator dependent anomalies are absent.

One can further add a local counterterm which is proportional to $dC \wedge dC \wedge dC$ with an appropriately normalized coefficient. This additional counterterm would further shift $u_4$ as

$$\Delta u_4 \propto C_2 \wedge C_2 \,, \tag{A.9}$$

which modifies the relation between the electric and magnetic symmetry.

# B  Circle Reduction and Massive States

In this appendix we briefly discuss the 5d perspective of the anomalous shift involving a non-trivial background for the center 1-form symmetries of the theory. The 5d perspective is indeed useful to support that the GSWS coupling comes from integrating out some massive states. It is believed that a 6d tensor multiplet defines a non-Abelian tensor when its tensor scalar $\langle \phi^i \rangle = 0$. This reduces to a non-Abelian gauge theory in 5d, which breaks into its Cartan $U(1)^i_{T_{6d}}$ when $\langle \phi^i \rangle \neq 0$. The W-bosons of this gauge theory are in general also charged under the vector fields $A^j$ of $G^j$, which can inherit the 1-form symmetries from 6d. The Chern–Simons coupling (2.36) between $U(1)^i_{T_{6d}}$ and $G^j$ then comes by integrating out these massive W-bosons [40, 41].

As a simplified example, let us consider a 5d theory constructed via M-theory on a local Calabi–Yau threefold that is the normal bundle of $\mathbb{F}_1 \cup \mathbb{F}_1 \equiv S_1 \cup S_2$ — two Hirzebruch-1 surfaces glued along a common $(-1)$-curve. In the singular limit, where we shrink the $\mathbb{P}^1$ fiber of both surfaces, it realizes an effective $SU(3)$ gauge theory with Chern–Simons level 0 [118]. Blowing-up one surface, say $S_1$, makes one set of W-bosons of the $SU(3)$ massive, leading to an effective $SU(2) \times U(1)$ theory, where the $SU(2)$ would have an $\mathbb{Z}_2$ 1-form center symmetry. This symmetry is explicitly broken by the massive W-bosons, as they have charge 1 under the $SU(2)$ Cartan, thus effectively being fundamental states. While these states are integrated out at low energies, their presence is in the coefficient of the Chern–Simons term $A_{U(1)} \wedge \text{Tr}(F_{SU(2)} \wedge F_{SU(2)})$. It is determined by the triple intersection number $S_1 \cdot S_2^2 = -1$, which yields the coupling (with normalization $\frac{1}{4}\text{Tr}(F^2)$ for the instanton density)

$$\frac{1}{2\pi} S \supset \int_{M_5} -A^1_{U(1)} \wedge \tfrac{1}{4}\text{Tr}(F_{SU(2)} \wedge F_{SU(2)}) \,. \tag{B.1}$$

This induces a non-trivial shift in 5d analogous to the one we analyze, see [17].

# C  Finite and Infinite Distance for Hirzebruch Surfaces

In this appendix we briefly comment on the different degenerations of Hirzebruch surfaces with focus on limits in which one of the curve volumes goes to zero.

The volumes of curves in the base manifold are controlled by the vacuum expectation values of the scalars in the tensor fields, whereas the overall volume $\mathcal{V}$ is determined by a hypermultiplet. In the following we set

$$\mathcal{V} = \tfrac{1}{2} \int_B J \wedge J = -\tfrac{1}{2} \Omega_{ij} \phi^i \phi^j = 1 \,, \tag{C.1}$$

where we decomposed $J = \phi^i \omega_i$ and $\omega_i$ the harmonic 2-forms dual to the 2-cycle classes. Under this condition the Hirzebruch surfaces $\mathbb{F}_n$ only have a single undetermined modulus:

$$2 = -n(\phi^1)^2 + 2\phi^1 \phi^2 \quad \Rightarrow \quad \phi^2 = \tfrac{1}{\phi^1} + \tfrac{n}{2} \phi^1 \,. \tag{C.2}$$

The metric on the moduli space is given by the kinetic matrix in the tensor sector,

$$g_{ij} = \phi_i \phi_j + \Omega_{ij} \,. \tag{C.3}$$

For the Hirzebruch surfaces this is given by

$$g_{ij} = \begin{pmatrix} \tfrac{1}{4} n^2 (\phi^1)^2 + \tfrac{1}{(\phi^1)^2} & -\tfrac{1}{2} n(\phi^1)^2 \\ -\tfrac{1}{2} n(\phi^1)^2 & (\phi^1)^2 \end{pmatrix} \,, \tag{C.4}$$

and one can measure the distance $s$ in field space by (see also [63, 119])

$$s(\phi_i^1, \phi_f^1) = \int_{\phi_i^1}^{\phi_f^1} \left( g_{ij} \, dj^i dj^j \right)^{1/2} = \int_{\phi_i^1}^{\phi_f^1} \frac{\sqrt{2} \, d\phi^1}{\phi^1} \,, \tag{C.5}$$

where, since there only is a single modulus, the distance is uniquely defined. The situation complicates in the presence of multiple moduli fields for which the distance is path dependent, and infinite distance points are points for which every possible path has infinite length.

We are interested in limits in which one of the curves goes to infinite volume. Denoting the self-intersection $(-n)$ curve by $C_n$ and the self-intersection $(0)$ curve by $C_0$ one has

$$\mathrm{vol}(C_n) = \int_{C_n} J = \phi_1 = \tfrac{1}{\phi^1} - \tfrac{n}{2} \phi^1 \,, \quad \mathrm{vol}(C_0) = \int_{C_0} J = \phi_2 = \phi^1 \,. \tag{C.6}$$

One sees that the point in moduli space where $C_n$ has zero volume is given by

$$\phi^1 = \left( \frac{2}{n} \right)^{1/2} \,, \tag{C.7}$$

at which $C_0$ remains of finite size. Moreover, this point is a finite distance away from a generic point on the tensor branch indicating that the SCFT limit is at finite distance. However, there is also an infinite distance limit for $j^1 \to 0^+$, in which case the volume of $C_0$ vanishes and the volume of $C_n$ diverges.

Note that even though one could be tempted to regard the latter as a little string theory limit, since the remaining intersection matrix has a single zero eigenvalue, this is not correct. In the LST limit gravity is decoupled but there is still a scale in the theory [23–25] which parametrizes the finite string tension. Therefore, one should regard the LST limit as a limit in which one simultaneously scales up the overall volume $\mathcal{V}$ while keeping the volume of the curve $C_0$ fixed. The limit described above with $\mathrm{vol}(C_0) \to 0$ also has a nice interpretation in terms of a dual heterotic string as discussed in [63, 94, 96].

## D  Explicit resolution

The explicit resolution of the $\mathbb{Z}_3$-torsion model on $\mathbb{F}_3$ can be computed via methods described in [120]. For that, we express the elliptic fibration in Tate form, $x^3 - y^2 + a_1 xyz + a_2 x^2 z^2 + a_3 yz^3 + a_4 xz^4 + a_6 z^6 = 0$, where $a_i$ is a section of $\overline{K}^{\otimes i}$. Then, to tune a $\mathbb{Z}_3$ Mordell–Weil group, one sets $a_2 = a_4 = a_6 = 0$ [60]. Furthermore, since on an $\mathbb{F}_3$, $a_1 = a_1' v$ and $a_3 = a_3' v$, we have

$$x^3 - y^2 + a_1' vxyz + a_3' vyz^3 = 0\,. \tag{D.1}$$

This fibration has two rational sections given by the intersection with $x = 0$:

$$\tau : (x, y, z) = (0, 0, 1) \quad \text{and} \quad \rho : (x, y, z) = (0, a_3' v, 1)\,. \tag{D.2}$$

They are conjugate to each other with respect to the $\mathbb{Z}_3$ group law.

The fibration also has singularities at $x = y = v = 0$ and $x = y = a_3' = 0$. To resolve these, we have to introduce two blow-ups for each singularity. Let us denote the blow-ups at $x = y = v = 0$ by $v_1$ and $v_2$, and those at $x = y = a_3' = 0$ by $u_1$ and $u_2$, then the resolved Calabi–Yau is the hypersurface

$$P := x^3\, u_1\, v_1 - y^2\, u_2\, v_2 + a_1'\, x\, y\, z\, v\, v_1\, v_2 + a_3'\, y\, z^3\, v = 0\,, \tag{D.3}$$

with the following Stanley–Reisner (SR) ideal for the ambient space:

$$\{u_1 z,\ u_2 z,\ v_1 z,\ v_2 z,\ u_1 v,\ u_2 v,\ u_1 y,\ v_1 y,\ u_1 v_1,\ u_2 v_1,\ xyz,\ xyv,\ xvv_2,\ a_3' xy,\ xu_0 u_2\}\,. \tag{D.4}$$

The exceptional divisors $v_{1,2}$ and $u_{1,2}$ are fibered over $\{v = 0\}$ and $\{a_3' = 0\}$ in the base, respectively.

Because of the SR-ideal, the section $\tau$ intersects the $\mathbb{P}^1$-fibers of the exceptional divisors $\{v_2 = 0\}$ and $\{u_2 = 0\}$. For the Mordell–Weil group law to be compatible with the fiber structure, $\rho$ mus then intersect the other two exceptional fibers, i.e., $\{v_1 = 0\}$ and $\{u_1 = 0\}$.

This means that the *Shioda-map* [90, 121], which is a divisor class $\varphi(s) = [s] - [\sigma_0] + \ldots$ that intersects none of the exceptional divisors, must take the form

$$\varphi(\tau) = [\tau] - [\sigma_0] + \frac{1}{3}(D_1^{(1)} + 2D_2^{(1)} + D_1^{(2)} + 2D_2^{(2)}) + \pi^{-1}(D_B),$$

$$\varphi(\rho) = [\rho] - [\sigma_0] + \frac{1}{3}(2D_1^{(1)} + D_2^{(1)} + 2D_1^{(2)} + D_2^{(2)}) + \pi^{-1}(D'_B),$$

(D.5)

with the divisor classes $D_{1,2}^{(1)} = [v_{1,2}]$ and $D_{1,2}^{(2)} = [u_{1,2}]$, and $\sigma_0$ the zero section. To infer the vertical parts $\pi^{-1}(D_B)$ and $\pi^{-1}(D'_B)$, we first note that the Shioda-map of a torsional section must be trivial in homology [61], implying

$$[\tau] = [\sigma_0] - \frac{1}{3}(D_1^{(1)} + 2D_2^{(1)} + D_1^{(2)} + 2D_2^{(2)}) - \pi^{-1}(D_B),$$

$$[\rho] = [\sigma_0] - \frac{1}{3}(2D_1^{(1)} + D_2^{(1)} + 2D_1^{(2)} + D_2^{(2)}) - \pi^{-1}(D'_B).$$

(D.6)

Then, in the ambient space, we can use the homology relation

$$[x] = 2\overline{\mathcal{K}} + 2[z] - D_1^{(1)} - D_2^{(1)} - D_1^{(2)} - D_2^{(2)},$$

(D.7)

with $\overline{\mathcal{K}}$ the pullback of the anti-canonical bundle of the base $\overline{K}$ to the ambient space, which is also fibered over $B = \mathbb{F}_3$. Since $\{x = 0\}$ restricts to the two $\mathbb{Z}_3$ sections $\tau$ and $\rho$, they must sum to the restriction of the above class to $\{P = 0\}$, with $[z]|_{\{P\}} = [\sigma_0]$ the zero section. This implies that $\pi^{-1}(D_B) + \pi^{-1}(D'_B) = -2\pi^{-1}(\overline{K})$. Because the sections are "symmetric" with respect to the base, we must have $\pi^{-1}(D_B) = \pi^{-1}(D'_B) = -\pi^{-1}(\overline{K})$.

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
