# Peer review of "The Fate of Discrete 1-Form Symmetries in 6d"

_SciPost Physics_

## Round 2 · Referee Report · Anonymous (Referee 1) · 2020-10-30

Report

This is an interesting paper in the context of higher-dimensional supersymmetric theories, which is currently a very active area.

This work presents some steps of an analysis of the higher form symmetries of six-dimensional theories from the perspective of the low energy effective action, and it points out an intriguing effect: there is a potential mixed anomaly between discrete global 1-form symmetry groups and the higher gauge transformation for anti-self dual tensors.

The referee recommends accepting this work for publication, provided the questions presented in the section "requested changes" below are addressed satisfactorily.

Requested changes

These are some questions/suggestions for the authors.

1.)
In 6d ordinary gauge anomaly cancellation strongly constrains the matter content of the theory.

The matter introduced to cancel the anomaly is typically charged with respect to the centre symmetry thus breaking it completely or to certain subgroups. For this reason only few examples of six-dimensional theories end up having interesting discrete 1-form centre symmetries.

As an example of this fact for 6d (1,0) theories the only gauge groups that can appear without matter are SU(3), SO(8), F4, E6, E7 and E8. These gauge groups are discussed in section 3 and are free of the dangerous fractionalization.

Similarly, this happens for the other non-higgsable models analyzed in section 3.

From the examples presented it seems that the constraints from 6d gauge anomaly cancellation are such that the effect observed around equations (3.9) and (3.10) is generic in the context of models with conventional matter (e.g. bi-fundamentals of various kinds): is there evidence in favor or against this idea? For instance, can the authors provide a more detailed analysis of the example in equation (3.18)?

2.)
In the same spirit of the above question: is there an alternative argument to argue that gauging the 0-form global symmetries of various types of conformal matter is not breaking the centre symmetry of the corresponding 0-form gauge groups? Same question also for gauging various subgroups of the E8 global symmetries of the E-string . One possible consistency check is given by the circle reduction of the corresponding models.

3.)
The theories considered in Section 5 are gravitational, hence should not have any global symmetry.

Gauging a 1-form centre symmetry does not necessarily produce a theory that does not have a higher symmetry, rather it typically give rise to a parent magnetic higher form symmetry.

For instance gauging the centre symmetry of a 4d SU(N) gauge theory one obtains PSU(N) which has a magnetic $\mathbb{Z}_N$ 1-form symmetry.

In 6d by a similar token gauging an electric 1-form symmetry might give rise to a theory with a magnetic 3-form symmetry.

If the theories discussed in section 5 are obtained by gauging 1-form symmetries, is there a way to argue that there are no other emergent magnetic 3-form higher symmetries from the gauging as required by a consistent gravitational model?

4.)
What is the matter content of the theories discussed in equation (5.17) of section 5.2? The intersection theory in equations (5.19) and (5.21) is reminiscent of the computations about the coefficients for the ordinary anomaly polynomial: can this remark be used to answer the referee's question 1.)?

---

## Round 2 · Referee Report · Anonymous (Referee 2) · 2021-5-26

Strengths

  1. Analyzes the intersection of two topics of interest: higher form symmetries (in this case, discrete symmetries), and higher dimensional theories.

  2. The analysis seems sound, with interesting results.

  3. The analysis and results will be useful for future research in these topics.

Weaknesses

The paper could be more clearly written. There are potentially confusing points in the presentation, analysis, and confusing comments. For example, the 4th sentence in the abstract refers to an "ambiguity". The next sentence calls it an "anomaly" and says that it is "eliminated" by charged strings. The first part of the next sentence says "However,..." and then goes back to explain why there was an anomaly in the first place. The second half of that sentence concludes "hence the symmetry is absent" but that conclusion is presumably meant to go back to the previous sentence about the charged strings. There are many such confusing circling around discussions in the writeup. As another example, in eqn. (2.5) and the following equations they sometimes say there are N_T tensors and sometimes N_T+1 and some of the equations are valid in one case and some in the other case. This has to do with whether or not they include the tensor of opposite signature from the gravity multiplet, but none of that is explained there.

Report

If I understand it correctly, I think that the summary of the conclusions could be stated much more succinctly than they do. As the other referee remarked, there are only a few cases where discrete 1-form center symmetries could possibly exist. As I understand it, the examples considered here have accidental, approximate 1-form center symmetries on the tensor branch. The paper shows that these center symmetries have a mixed anomaly with the 2-form gauge symmetry which prevents turning on non-trivial backgrounds for the center symmetries. Rather than curing the anomaly, the observation here that BPS strings are charged show that the center symmetry is actually explicitly broken, and is at best an accidental symmetry away from the origin of the tensor branch. It's nice that the anomaly of the accidental symmetry knows to encode the fact that there is actually no such symmetry at the origin. There is of course also no such global symmetry when including gravity.

I believe that the paper has interesting results, and it should be published.

Requested changes

I do not have specific requested changes, only the general comment that I think that the paper could have been written more clearly. The authors should address the other referee's comments and make a pass through the paper to see if they can improve the presentation.

---

## Editorial Decision

resubmitted